# *Zfp296* knockout enhances chromatin accessibility and induces a unique state of pluripotency in embryonic stem cells

Satsuki Miyazaki[1], Hiroyuki Yamano[2], Daisuke Motooka[2], Fumi Tashiro[3], Takumi Matsuura [1,4], Tatsushi Miyazaki[1] & Jun-ichi Miyazaki [3✉]

The *Zfp296* gene encodes a zinc finger-type protein. Its expression is high in mouse embryonic stem cells (ESCs) but rapidly decreases following differentiation. *Zfp296*-knockout (KO) ESCs grew as flat colonies, which were reverted to rounded colonies by exogenous expression of *Zfp296*. KO ESCs could not form teratomas when transplanted into mice but could efficiently contribute to germline-competent chimeric mice following blastocyst injection. Transcriptome analysis revealed that *Zfp296* deficiency up- and down-regulates a distinct group of genes, among which *Dppa3, Otx2,* and *Pou3f1* were markedly downregulated. Chromatin immunoprecipitation sequencing demonstrated that ZFP296 binding is predominantly seen in the vicinity of the transcription start sites (TSSs) of a number of genes, and ZFP296 was suggested to negatively regulate transcription. Consistently, chromatin accessibility assay clearly showed that ZFP296 binding reduces the accessibility of the TSS regions of target genes. *Zfp296*-KO ESCs showed increased histone H3K9 di- and tri-methylation. Co-immunoprecipitation analyses revealed interaction of ZFP296 with G9a and GLP. These results show that ZFP296 plays essential roles in maintaining the global epigenetic state of ESCs through multiple mechanisms including activation of *Dppa3*, attenuation of chromatin accessibility, and repression of H3K9 methylation, but that *Zfp296*-KO ESCs retain a unique state of pluripotency while lacking the teratoma-forming ability.

[1] Division of Stem Cell Regulation Research, Center for Medical Research and Education, Osaka University Graduate School of Medicine, 2-2 Yamadaoka, Suita, Osaka 565-0871, Japan. [2] Genome Information Research Center, Research Institute for Microbial Diseases, Osaka University, 3-1 Yamadaoka, Suita, Osaka 565-0871, Japan. [3] The Institute of Scientific and Industrial Research (SANKEN), Osaka University, 8-1 Mihogaoka, Ibaraki, Osaka 567-0047, Japan. [4] Present address: Toray Industries, Inc., Tokyo, Japan. ✉email: jimiyaza@nutri.med.osaka-u.ac.jp

In an attempt to elucidate the key stemness factors related in the pluripotency of mouse embryonic stem cells (ESCs), expressed sequence tag counts were compared between ESCs and somatic tissues with digital differential display[1–4]. The *Zfp296* gene was included in the top 20 genes with the highest enrichment in ESCs using digital differential display[1]. The expression of this gene is very high in undifferentiated ESCs but rapidly decreases following the induction of differentiation. Thus, *Zfp296* has been used as a marker for undifferentiated ES and reprogrammed cells[5,6]. *Zfp296* is expressed in testis, bone marrow, etc. in adult mice[7]. Its expression is also seen in early mouse embryos including the inner cell mass of the blastocysts[8]. Such expression patterns suggest that it may function specifically in stem and progenitor cells.

The mouse *Zfp296* gene encodes a protein of 445 amino acid residues containing 6 zinc finger (ZF) domains[7]. This gene is conserved in human, gorilla, dog, cow, rat, etc. We also found that the C-terminal ZF domains of mouse ZFP296 protein are highly homologous to those of mouse CTIP2 (BCL11B) in amino acid sequences. The *Zfp296* gene was identified as a proviral integration site in BHX2 mice with retrovirally induced myeloid leukemia[9,10]. *ZNF342*, the human homologue of *Zfp296*, has 75% nucleotide homology and was also reported to be responsible for pediatric acute myeloid leukemia (AML) caused by chromosomal rearrangements[11]. Transcription of *ZNF342* was downregulated by hypermethylation of CpG island in oligodendroglioma[12] and prostatic carcinoma[13]. Thus, *Zfp296* is assumed to play important roles in cell proliferation, survival, differentiation, and carcinogenesis. Among 23 genes specifically expressed in ESCs, the *Zfp296* gene showed the highest ability to enhance the production of induced pluripotent stem (iPS) cells, and ZFP296 was shown to activate the *Oct3/4* gene via its germ cell-specific conserved region 4 (CR4) containing the distal enhancer[8]. It was reported that ZFP296 binds to the DNA-binding domain of KLF4 and that the *Lefty1* promoter activation by KLF4 was repressed by ZFP296, suggesting the role of ZFP296 as a negative regulator[14]. These studies suggested that ZFP296 plays important roles in the maintenance of pluripotency of ESCs as a transcription regulator.

We recently reported that a *Zfp296* deficiency in mice impairs germ-cell development and embryonic growth[15]. ZFP296 protein was intracellularly localized to heterochromatin. In *Zfp296*-deficient mouse embryos, we observed a global increase in H3K9 methylation in a developmental stage-dependent manner and an increase in the H3K9me3 levels at major satellite repeats (MSR). Our results showed that ZFP296 is a component of heterochromatin and negatively regulates H3K9 methylation. Following our study, two other groups reported the roles of ZFP296. Using genome-wide CRISPR screening in ESCs, *Zfp296* was identified as a gene for acquisition of primordial germ cell (PGC) fate, and abrogation of the *Zfp296* gene was shown to result in widespread inhibition of WNT pathway factors in PGC-like cells, leading to failure to activate germline genes and consequently loss of germ cell identity[16]. Their study indicates ZFP296 to be a key component of an expanded PGC gene regulatory network. ZFP296 was also identified as an ESC-specific NuRD (nucleosome remodeling and deacetylase) complex interactor which regulates genome-wide NuRD binding and cellular differentiation[17]. Because the NuRD complex is a protein complex coupling chromatin remodeling ATPase and chromatin deacetylation enzymatic functions[18], ZFP296 is thought to function as a epigenetic regulator.

The present study aimed to elucidate the roles of ZFP296 in the maintenance of self-renewal and pluripotency of ESCs. We generated *Zfp296*-deficient and -rescue ESC lines and performed detailed analyses of these cell lines including their differentiation potential in vivo and in vitro and gene expression patterns. We also examined the ZFP296-binding loci in the whole genome by the chromatin immunoprecipitation (ChIP) sequencing and the effects of ZFP296 binding on chromatin accessibility by the assay for transposase-accessible chromatin (ATAC) sequencing[19]. Based on the results obtained in this study, we discuss the roles of ZFP296 in ESC maintenance through transcriptional and epigenetic mechanisms.

## Results

**Establishment of *Zfp296*-deficient ESCs.** The expression of *Zfp296* during ESC differentiation into trophectoderm was examined using the ZHBTc4 ESC line, in which expression of *Pou5f1* encoding Oct3/4 can be downregulated by tetracycline[20]. As shown in Supplementary Fig. 1a, *Zfp296* expression rapidly decreased, suggesting that *Zfp296* is downstream of Oct3/4. To generate *Zfp296*-deficient ESCs, we used a gene-targeting vector in which the IRES-βgeo cassette was inserted into the *Zfp296* gene (Supplementary Fig. 1b). This targeting vector was designed to disrupt all of the six ZF domains of ZFP296[15]. Two of the resulting *Zfp296*+/− clones were further subjected to double knockout (KO) using another gene-targeting vector in which the IRES-puro cassette was inserted in place of the IRES-βgeo cassette (Supplementary Fig. 1b), and several *Zfp296*−/− clones were obtained. We chose two *Zfp296*−/− clones, KO #98 and #156, derived from different *Zfp296*+/− clones for further analysis. Disruption of both alleles of the *Zfp296* gene in these clones was confirmed by RT-PCR and western blot analysis (Supplementary Fig. 1c, d). We also performed karyotype analysis for these *Zfp296*−/− clones. The results showed that most of the cells from KO #98 and #156 clones were karyotypically normal (7/7 and 11/14, respectively).

As we thought it important to confirm that the phenotypes seen in the *Zfp296*−/− clones are due to *Zfp296* deficiency, we rescued *Zfp296* expression in *Zfp296*−/− ESC clones (KO #98 and #156) by stably introducing a plasmid vector expressing *Zfp296* cDNA (pCAG-Zfp296-IZ). We obtained transfectant clones, Rescue #22 from KO #98 and Rescue #10 from KO #156 (Supplementary Fig. 1c, d). KO #98 cells were also stably transfected with an EGFP (enhanced green fluorescent protein) expression plasmid, pCAG-EGFP-IZ, and the resulting EGFP-expressing *Zfp296*−/− clone (KO-EGFP #26) was used as a control.

**Characterization of *Zfp296*-deficient ESCs.** ESCs grown in conventional serum/leukemia inhibitory factor (LIF) medium without feeder cells are heterogeneous in their colony morphology. Previously, it was reported that colonies mainly composed of *Rex1*-positive ESCs exhibited compacted morphology, but those of *Rex1*-/*Oct3/4*+ ESCs showed flat morphology[21]. They suggested that undifferentiated ESCs contain subpopulations corresponding not only to inner cell mass (ICM) but also to epiblast or primitive ectoderm. Thus, the colony morphology has been considered to reflect the differentiation status of ESC subpopulations. Therefore, we were interested in the effects of *Zfp296* deficiency on the colony morphology of ESCs.

We compared the colony morphology among KO #98 and #156, Rescue #22 and #10, and wild-type (WT) ESCs. Interestingly, *Zfp296*-deficient ESC colonies showed completely flat morphology, which is distinct from round compacted colony morphology of undifferentiated mouse ESCs (Fig. 1a). Exogenous *Zfp296* expression clearly reverted the flat colony morphology to tightly compacted one. These results suggested that *Zfp296* has a critical role in the maintenance of pluripotent state of ESCs. Next, we examined the colony morphology of KO #98 ESCs expressing ZFP296 mutants lacking the 2nd-3rd (ΔZF2-3), 4th-6th (ΔZF4-6), or 6th (ΔZF6) ZF domains[15]. As shown in Supplementary

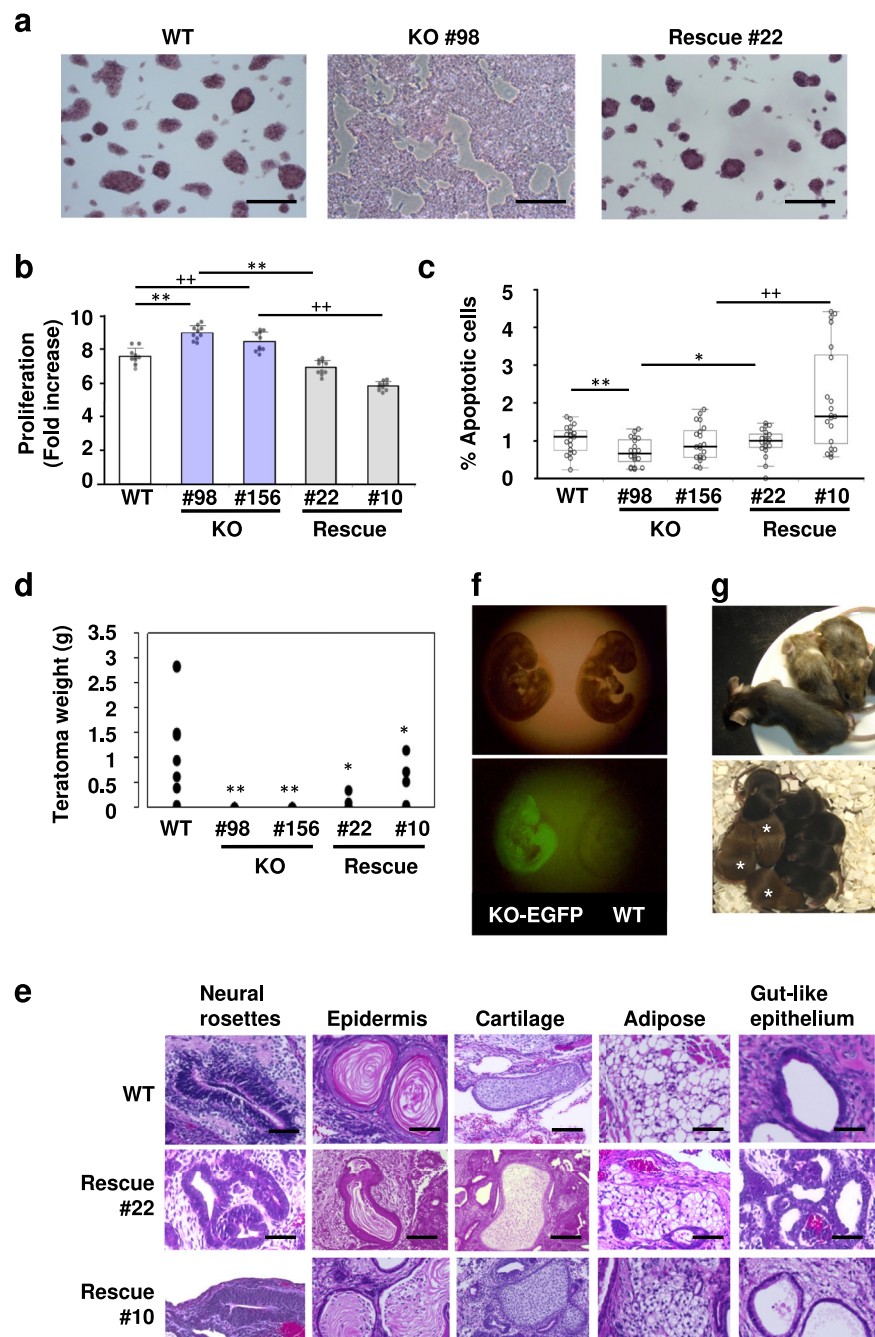

**Fig. 1 Effects of the presence or absence of *Zfp296* expression on ESC colony morphology, growth, apoptosis, teratoma formation, and chimeric mouse formation. a** Colony morphology. Wild-type (WT), *Zfp296⁻/⁻* #98 (KO), and *Zfp296⁻/⁻*(CAG-Zfp296) #22 (Rescue) ESCs were cultured and stained for alkaline phosphatase. Scale bars, 300 μm. **b** Cell proliferation assay. Cell proliferation between 24 and 72 h of culture was measured using a cell counting kit. The proliferation rate of KO #98 and #156 ESCs was significantly higher than that of WT, Rescue #22, and #10 ESCs. Values are expressed as means ± SD ($n = 10$ each). Difference from KO #98 cells: $**P < 0.01$ by Student's $t$-test. Difference from KO #156 cells: $++P < 0.01$ by Student's $t$-test. **c** Apoptosis assay. The percentage of cleaved caspase 3 (CCP3)-positive apoptotic cells was calculated for 20 areas selected at random (> 300 nuclei per area). The percentage of CCP3-positive cells in KO #98 cells was significantly lower than that in WT or Rescue #22 cells. Values are expressed as means ± SD. Difference from KO ESCs: $*P < 0.05$, $**P < 0.01$, $++P < 0.01$ by Student's $t$-test. **d** Teratoma formation. WT, KO #98 and #156, and Rescue #22 and #10 ESCs ($n = 8$ each) were injected subcutaneously into immunodeficient or histocompatible mice. Four weeks later, these mice were sacrificed, and the tumors were weighed. Neither KO #98 nor #156 ESCs formed recognizable teratomas. Difference from WT cells: $*P < 0.05$, $**P < 0.01$ by Student's $t$-test. **e** Histological analyses of the teratomas derived from WT, Rescue #22, and #10 ESCs. No histological differences were recognized among these teratomas. Scale bars, 50 μm. **f**, **g** Blastocyst injection. WT and KO-EGFP #26 ESCs, which are EGFP-expressing *Zfp296*-deficient cells derived from KO #98 cells, were injected into C57BL/6 J blastocysts, which were then transferred into the uteri of pseudopregnant female mice. Broad contribution of KO-EGFP cells to the resulting fetus was confirmed at day 10 p.c. by EGFP fluorescence (left fetus) (**f**, lower panel). Chimeric mice were born (**g**, upper panel). When male chimeras were mated to C57BL/6 J females, the progeny included mice with agouti coat color (**g**, lower panel; marked by asterisk), indicating germline transmission of KO-EGFP ESCs. Genotyping of these agouti progeny showed that they harbor either of the targeted alleles of *Zfp296*.

Fig. 2, both ΔZF2-3 and ΔZF6 could revert flat colony morphology of KO ESCs to compacted one, but ΔZF4-6 could not, suggesting the functional importance of the 4th and/or 5th ZF domains of ZFP296.

We measured the cell growth of WT, KO, and Rescue ESCs. *Zfp296* deficiency increased the cell proliferation, and *Zfp296* overexpression reduced it (Fig. 1b). The effect of *Zfp296* deficiency on apoptosis was also examined by immunostaining cleaved caspase 3 (CCP3) (Fig. 1c). KO #98 ESCs showed approximately 30% reduction in percent CCP3-positive cells, compared with WT and Rescue #22 ESCs. Thus, *Zfp296* expression in ESCs reduced the cell proliferation and increased the frequency of apoptotic cells.

**In vivo differentiation capacity of *Zfp296*-deficient ESCs.** We examined the in vivo differentiation capacity of these cells by teratoma formation assay. Interestingly, KO #98 and #156 cells did not form any recognizable tumors. In contrast, Rescue #22 and #10 cells produced tumors, although their sizes tended to be smaller than those from WT ESCs (Fig. 1d). Histological analysis of these teratomas revealed differentiation of WT and Rescue ESCs to all three germ layer lineages, such as neural rosettes (ectoderm), epidermis (ectoderm), adipose tissue (mesoderm), cartilage (mesoderm), and gut-like epithelium (endoderm) (Fig. 1e). Considering that KO ESCs exhibited an even higher proliferation rate than WT ESCs in vitro, these results suggest that *Zfp296* deficiency severely affects the in vivo differentiation capacity of ESCs.

We next examined the ability of KO ESCs to generate chimeric mice. KO-EGFP #26 ESCs were injected into C57BL/6 J blastocysts, which were transferred into the uteri of pseudopregnant female mice. To our surprise, broad contribution of KO-EGFP cells to the resulting fetus was confirmed at day 10 post coitum (Fig. 1f). Furthermore, chimeric mice were born (Fig. 1g, upper panel). When male chimeras were mated to C57BL/6 J females, the resulting progeny included mice of agouti coat color, indicating germline transmission of KO-EGFP cells (Fig. 1g, lower panel). In fact, these agouti progeny were confirmed to have either of the two KO alleles of *Zfp296*. Thus, *Zfp296*-deficient ESCs retain the ability to generate germline-transmitting chimeras, although they have lost the ability to form teratomas.

**Expression of stem-cell and differentiation marker genes.** To examine the effects of the presence or absence of ZFP296 on the expression of stem-cell and differentiation marker genes, WT, KO #98 and #156, and Rescue #22 and #10 ESCs were cultured with serum/LIF, and expression of these genes was monitored by quantitative RT-PCR analysis (Supplementary Fig. 3a). We observed a significant decrease of the expression of *T* (*Bra*), a mesoderm marker, and a significant increase of the expression of *Sox1*, an ectoderm marker, and *Gata4*, an endoderm marker, in two KO ESC clones, compared with WT and Rescue ESCs. However, none of the other stem-cell markers or differentiation markers tested were considerably affected.

We then induced differentiation of WT, KO #98 and #156, and Rescue #22 and #10 ESCs by LIF withdrawal and compared expression of stem-cell and differentiation marker genes among these cells on day 5 and 8 (Supplementary Fig. 3a). The expression levels of stem-cell marker genes, such as *Pou5f1*, *Nanog*, and *Zfp42* (*Rex1*), were kept high in KO cells till day 8. *Gata4* expression was lower and *Sox1* expression was higher in KO cells at day 5, but these differences were not prominent at day 8. Otherwise, we could not observe any considerable differences in the expression of stem-cell and differentiation marker genes among these ESCs. In suspension culture without LIF, WT, KO

#98, and Rescue #22 ESCs formed embryoid bodies (EBs), which started beating after day 6 (Supplementary Fig. 3b). The percentage of beating EBs was significantly lower in Rescue ESCs, compared with WT and KO ESCs during the observation period, suggesting that persistent expression of *Zfp296* may exert inhibitory effects on mesodermal differentiation.

**Effects of the presence or absence of ZFP296 on the transcriptome.** The above studies demonstrated that *Zfp296*-KO ESCs grew as flat colonies and could not form teratomas, but could contribute to germline-competent chimeric mice. To analyze the molecular mechanisms causing these unique phenotypes of *Zfp296*-KO ESCs, we compared gene expression profiles among WT, KO #98 and #156, and Rescue #22 and #10 ESCs by RNA-sequencing (RNA-seq). The genes whose expression was the most severely affected are shown in Fig. 2a. Genes markedly downregulated in KO cells included *Gbp2*, *Plac8*, *Myl9*, *Pou3f1*, *Dppa3*, *Otx2*, *Pim2*, etc. (Fig. 2a, left panel; Fig. 2b). Interestingly, these genes were reported to be highly induced in epiblast-like cells (EpiLCs) which were derived from ESCs in vitro and considered to closely resemble early-stage epiblast[22]. Furthermore, genes markedly upregulated in KO cells included *Napsa*, *Ntn1*, *Cdx1*, *Aard*, *Nkx6-3*, *Mras*, *Ncam1*, etc. (Fig. 2a, right panel; Fig. 2b), which are strongly repressed in EpiLCs[22]. Quantitative RT-PCR analysis was performed for some of the affected genes among WT, KO #98, and Rescue #22 ESCs (Supplementary Fig. 4a) and also among WT, KO #156, and Rescue #10 ESCs (Supplementary Fig. 4b), and the results showed gene expression changes consistent with the RNA-seq analysis.

**Reduced expression of *Dppa3* in *Zfp296*-KO ESCs.** Among the genes downregulated upon *Zfp296* KO, we focused on the *Dppa3* (*Developmental pluripotency-associated protein 3*, also known as *Pgc7* or *Stella*) gene. Its expression is very high in naïve ESCs but lost in epiblast stem cells (EpiSCs)[23], and it is involved in global DNA demethylation in ESCs[24]. We examined *Dppa3* expression in WT, KO #98, and Rescue #22 ESCs by immunofluorescence microscopy (Fig. 3a). DPPA3 was clearly detected in WT but only weakly in KO ESCs, whose *Dppa3* expression was restored by exogenous *Zfp296* expression. These results were also confirmed by western blotting (Fig. 3b).

Silencing of the *Dppa3* gene in EpiSCs is accompanied by the CpG hypermethylation of its promoter region[25]. Therefore, we examined DNA methylation levels of the *Dppa3* promoter region by bisulfite sequencing (Fig. 3c). The methylation level of this region was low in WT ESCs (26.1%), but this region became highly methylated in KO #98 and #156 ESCs (93.9% and 90.5%, respectively). The methylation level of this region was reduced in Rescue #22 and #10 ESCs (62.2% and 65.4%, respectively). Considering that the *Dppa3* expression in Rescue ESCs was comparable to that in WT ESCs, DNA methylation levels of the promoter region in Rescue ESCs seemed high. However, we do not know the exact reason for this inconsistency. The above results indicated that the expression of *Dppa3* was epigenetically regulated by the presence or absence of ZFP296. DPPA3 has been reported to inhibit the maintenance DNA methylation[26,27] and induce global DNA demethylation[24]. Therefore, severe repression of *Dppa3* observed in *Zfp296*-KO ESCs is assumed to affect global DNA methylation levels. We examined the levels of 5mC and 5hmC among WT, KO #98 and #156, and Rescue #22 and #10 ESCs. As shown in Fig. 3d, 5hmC levels of KO ESCs were consistently higher than those of WT and Rescue ESCs. However, 5mC levels were not different among these cells. It is not known why *Zfp296*-KO cells did not show an increase in 5mC levels in spite of a marked reduction in *Dppa3* expression, but other

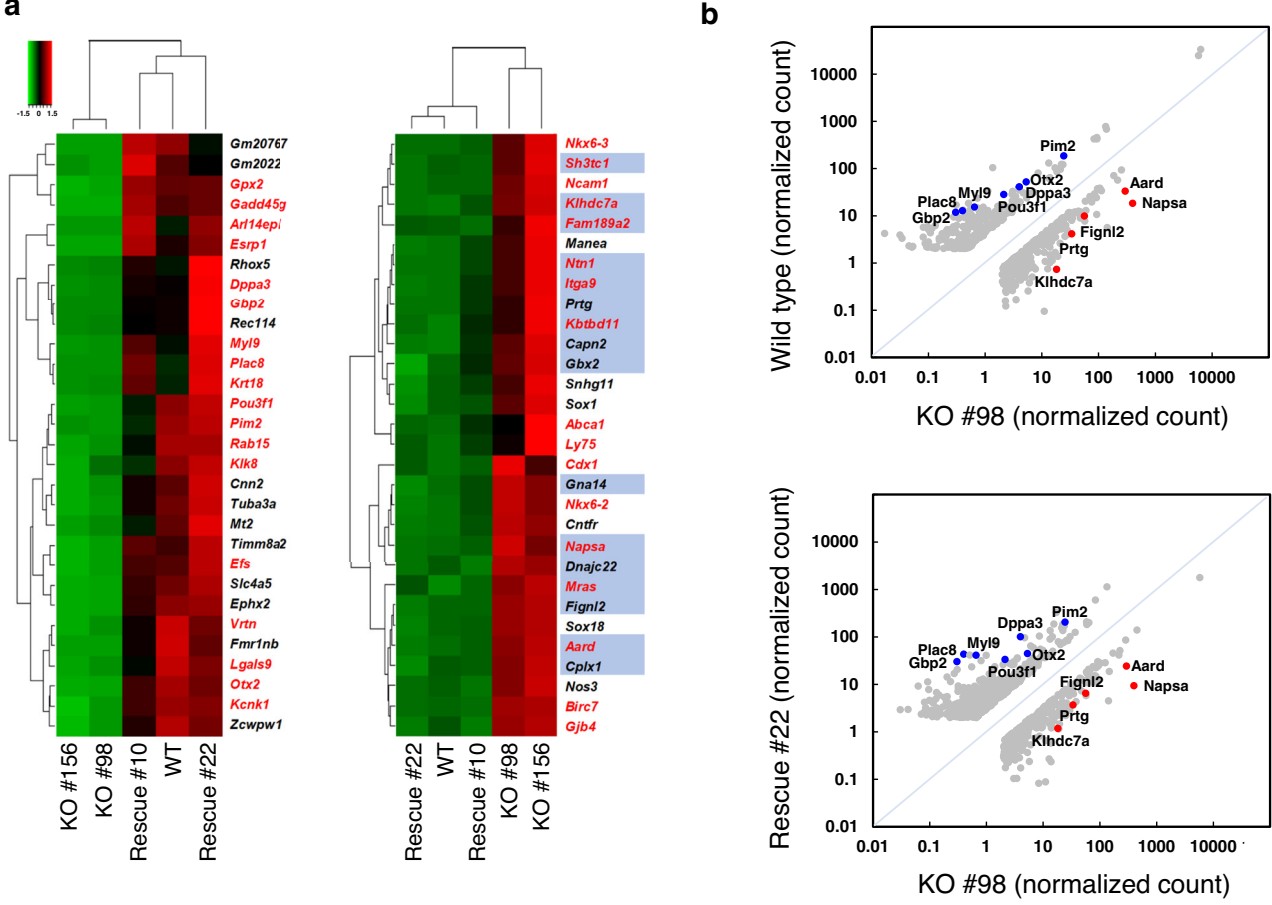

**Fig. 2 Effects of *Zfp296* deficiency and overexpression on the transcriptome of ESCs. a** Heatmap showing Z-score normalized transcript levels of the genes differentially expressed among WT, KO #98 and #156, and Rescue #22 and #10 ESCs. Red and green colors represent up- and down-regulation, respectively. The left and right panels show the top 30 genes most highly down- and up-regulated upon *Zfp296* KO, respectively (KO vs. WT ESCs, *P* < 0.05). Genes with normalized read counts less than 8 in all the ESCs and noncoding genes were omitted. Dendrograms show hierarchical clustering analysis of the five ESC lines and the 30 genes in each panel. ZFP296 binding peaks from ChIP-seq were found in the vicinity of 16 genes (highlighted in blue) out of the 30 genes upregulated upon *Zfp296* KO. Genes highly induced and strongly repressed in EpiLCs[22] are shown in red in the left and right panels, respectively. **b** Scatterplots of RNA-seq TMM normalized read counts of the genes that showed more than 3-fold (WT > KO #98 ESCs) or 2.5-fold (WT < KO #98 ESCs) difference (upper panel) and those that showed more than 3-fold difference (Rescue #22 vs. KO #98 ESCs; lower panel). Genes with normalized read counts less than 2 in all the ESCs were omitted. Blue and red dots represent some of the genes markedly down- and up-regulated in KO ESCs, respectively.

factors involved in cytosine methylation or demethylation might have been affected by *Zfp296* KO.

**Genome-wide analysis of ZFP296 binding loci**. Our RNA-seq analysis showed that the presence or absence of ZFP296 profoundly affects the gene expression patterns of ESCs. ZFP296 belongs to the C2H2-type ZF family and was proposed to function as a transcription factor[8]. Therefore, it seemed important to map the ZFP296 binding sites across the whole mouse genome of ESCs. We applied chromatin immunoprecipitation followed by high-throughput DNA sequencing (ChIP-seq) to genome-wide mapping of the ZFP296 binding sites. We introduced a plasmid expressing Ty1-tagged ZFP296 into KO #98 ESCs, resulting in *Zfp296*−/−(CAG-Ty1-Zfp296) ESCs, which showed compacted colony morphology. This cell line was used for ChIP assay with anti-Ty1 antibody followed by sequencing. Approximately 3000 ZFP296 binding peaks were identified, and their distribution along the genome was determined. We found that ZFP296 binding sites were selectively enriched in promoter regions and gene bodies (Fig. 4a), especially within 1 kb up- and 1.5 kb downstream of transcription start sites (TSS) (Fig. 4b), supporting the

role of ZFP296 in transcriptional regulation. Furthermore, we analyzed Gene Ontology (GO) biological processes[28] enriched in the genes located near ZFP296 binding sites and found significant enrichment of the genes related to "epigenetic regulation of gene expression" (Fig. 4c).

We next computationally identified consensus DNA binding motifs for ZFP296 in the retrieved sequences of the ZFP296 binding peaks using MEME[29] and DREME softwares[30] (Fig. 4d). The results provided several potential consensus binding motifs, such as GGA/TCA and GGCGTCCC/A. Interestingly, two long consensus sequences were obtained using DREME, and we found that one of them (29 bp long) exactly corresponds to a part of the 234 bp MSR sequences[31]. MSR sequences are located in pericentromeric heterochromatin[32]. Major satellites from different chromosomes are known to form clusters associated with heterochromatin protein 1α (HP1α). Our immunofluorescence microscopic analysis demonstrated that ZFP296 is localized to the DAPI (4',6-diamidino-2-phenylindole)-dense heterochromatin in E9.75 embryos, 293 T cells[15], and ESCs (Fig. 4e), which is consistent with the binding of ZFP296 within the MSR sequences. Thus, our ChIP assay revealed the genome-wide binding

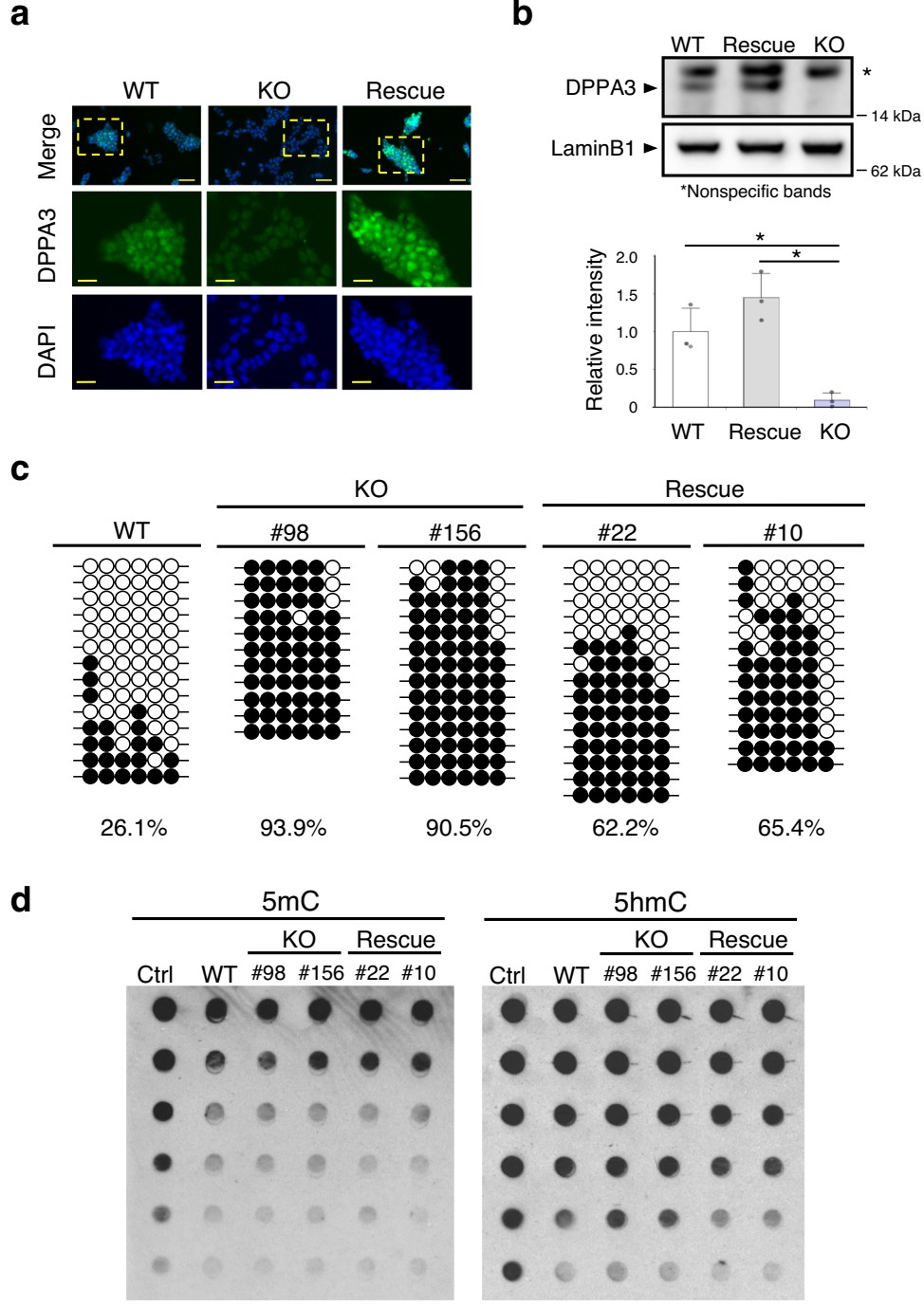

**Fig. 3 Reduced expression of *Dppa3* in *Zfp296*-KO ESCs. a** Immunofuorescence analysis of DPPA3 expression in WT, *Zfp296*-KO #98, and Rescue #22 ESCs. Nuclei were counterstained with DAPI. Scale bars, 50 μm (upper panels), 25 μm (middle and lower panels). **b** Western blot analysis of DPPA3 expression in WT, Rescue #22, and KO #98 ESCs (upper panel). Signal intensity of each DPPA3 band was measured relative to that of the Lamin B1 band. The lower panel shows the DPPA3 levels expressed relative to those in WT ESCs and as means ± SD (*n* = 3). *P < 0.05 by Tukey's test. **c** Analysis of the DNA methylation levels of the *Dppa3* promoter. Genomic DNA was isolated from WT, KO #98 and #156, and Rescue #22 and #10 ESCs. The CpG islands in the upstream region of the *Dppa3* gene were analyzed for cytosine methylation. Filled and open circles represent methylated and unmethylated CpGs, respectively. The percentage of methylated CpG sites is shown under each panel. **d** Analysis of 5mC and 5hmC levels by dot blot assay. Control 5mC and 5hmC DNA and genomic DNA from WT, KO #98 and #156, and Rescue #22 and #10 ESCs were denatured. Two-fold serially diluted DNAs were spotted onto membranes. Antibody against 5mC (left panel) or 5hmC (right panel) was utilized for detection.

properties of ZFP296 and also confirmed its binding to heterochromatin.

**ZFP296 binding sites and gene expression**. To relate ZFP296 binding to gene expression dynamics, we searched for ZFP296 binding peaks from 3 kb upstream of the TSS to 3 kb downstream of the transcription termination site for the genes highly upregulated and downregulated upon *Zfp296* KO in our RNA-seq analysis (Fig. 2a). We found ZFP296 binding in the vicinity of 16 genes out of the top 30 upregulated genes but did not out of the top 30

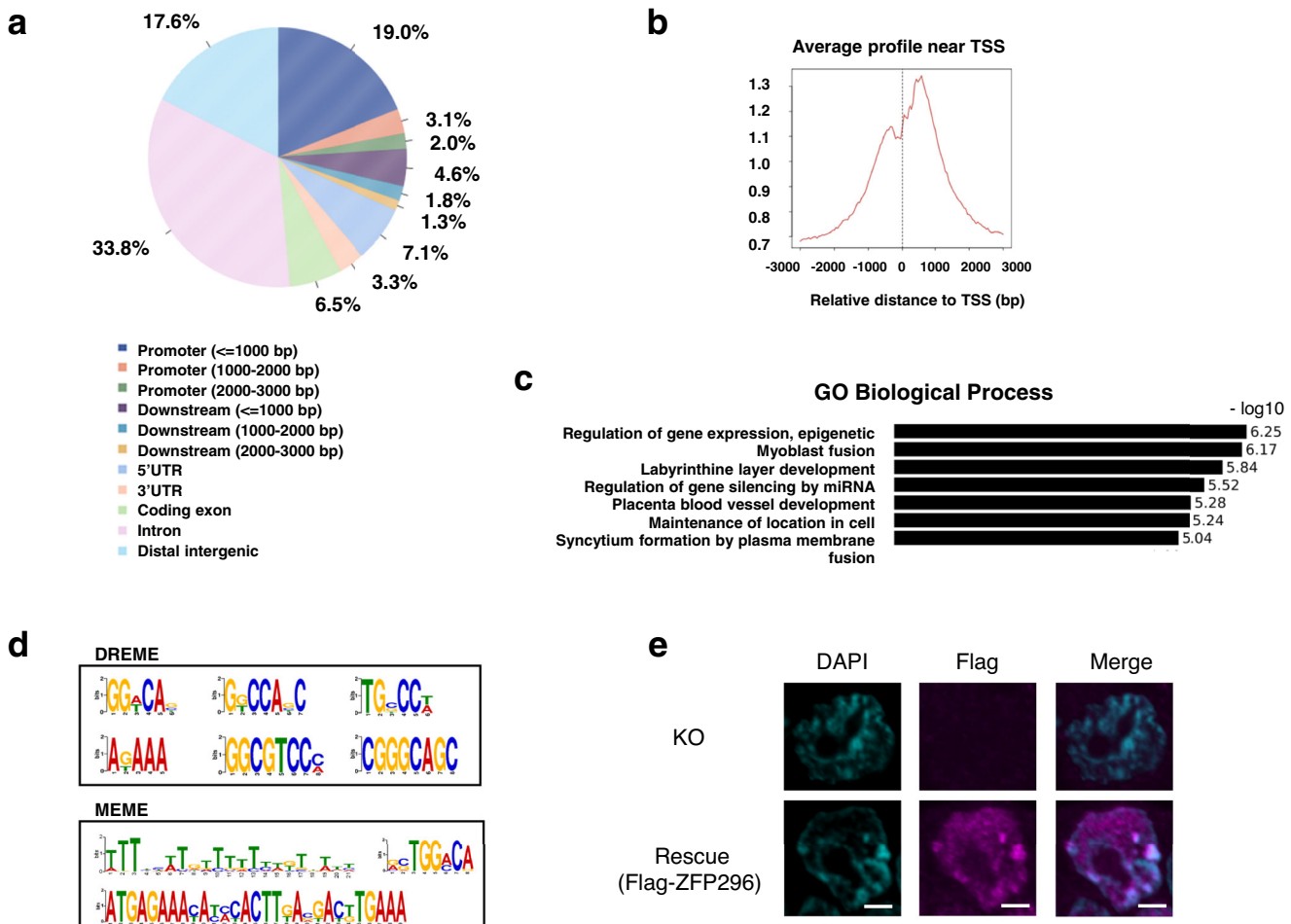

**Fig. 4 Analysis of ZFP296 binding sites by ZFP296 ChIP-seq. a** Pie chart showing the distribution of ZFP296 ChIP-seq peaks across genomic regions. **b** Plot showing the distribution of ChIP-seq peaks relative to known TSS. **c** Gene ontology (GO) biological processes enriched in the genes located near ZFP296-binding loci. **d** Most abundant DNA sequence motifs identified in ZFP296 ChIP-seq peaks. Consensus DNA binding motifs were computationally identified in the retrieved sequences of the ZFP296 binding peaks. **e** Localization of ZFP296 to heterochromatin. KO #98 and *Zfp296−/−*(CAG-Flag-Zfp296) cells (Rescue) were stained with anti-Flag-tag antibody followed by Alexa Fluor-labeled second antibody. Cells were counterstained with DAPI and observed by confocal fluorescence microscopy. Scale bars, 2 μm.

downregulated genes, suggesting that ZFP296 binding negatively regulates transcription. Examples of these analyses are shown in Fig. 5c, d. *Plac8*, *Dppa3*, and *Otx2* were strongly downregulated upon *Zfp296* KO, but our ChIP-seq analysis did not detect significant ZFP296 binding in the vicinity of these genes. By contrast, we found ZFP296 binding peaks in the vicinity of the genes, such as *Klhdc7a*, *Aard*, and *Napsa*, whose expression was highly upregulated upon *Zfp296* KO. ZFP296 binding peaks were often found in the promoter region or in the 1st intron of these upregulated genes (Figs. 2a and 4a, b). However, we could not find significant ZFP296-binding peaks in the vicinity of some of the upregulated genes, such as *Nkx6-3*, *Cdx1*, *Sox18*, and *Sox1* (Fig. 2a). It is likely that these genes are regulated by ZFP296 through indirect mechanisms, as is probably the case for the genes downregulated upon *Zfp296* KO.

**Effects of ZFP296 binding on chromatin accessibility.** To explore the molecular mechanism underlying transcriptional repression by ZFP296, we performed genome-wide analysis of chromatin accessibility (ATAC-seq; see Methods)[19]. We compared ATAC-seq datasets obtained from WT, KO #98, and Rescue #22 ESCs and characterized global alterations in chromatin accessibility. Accessible regions were primarily located around TSS (Fig. 5a) and also around the ZFP296 binding peaks

(Fig. 5b). In WT ESCs, the central portions of the accessible peaks around the ZFP296 binding peaks appear collapsed, representing a flattened peak shape. Interestingly, in KO ESCs, these portions became much more prominent as peaks but were almost completely reverted in Rescue ESCs, indicating that ZFP296 reduces chromatin accessibility around its binding sites.

We integrated the ATAC-seq data with the ZFP296 ChIP-seq and RNA-seq data to evaluate whether the chromatin accessibility changes correlate with ZFP296 binding and transcription levels. For the *Plac8*, *Dppa3*, and *Otx2* genes (Fig. 5c), significant binding of ZFP296 was not observed around the TSS or in the gene bodies, and the expression of these genes was severely reduced in KO ESCs compared with WT and Rescue ESCs. ATAC-seq peaks were observed around the TSS of these genes, but not of *Plac8* in KO ESCs. For the *Klhdc7a*, *Aard*, and *Napsa* genes (Fig. 5d), ZFP296 binding peaks were found near the TSS, and the expression of these genes was high in KO ESCs but very low in WT and Rescue ESCs. Consistent with these transcriptional changes, chromatin accessibility around their TSS regions was very low in WT and Rescue ESCs but was dramatically enhanced in KO ESCs.

We next examined the degree of overlap of ATAC-seq peaks among WT, KO, and Rescue ESCs. We first analyzed those peaks

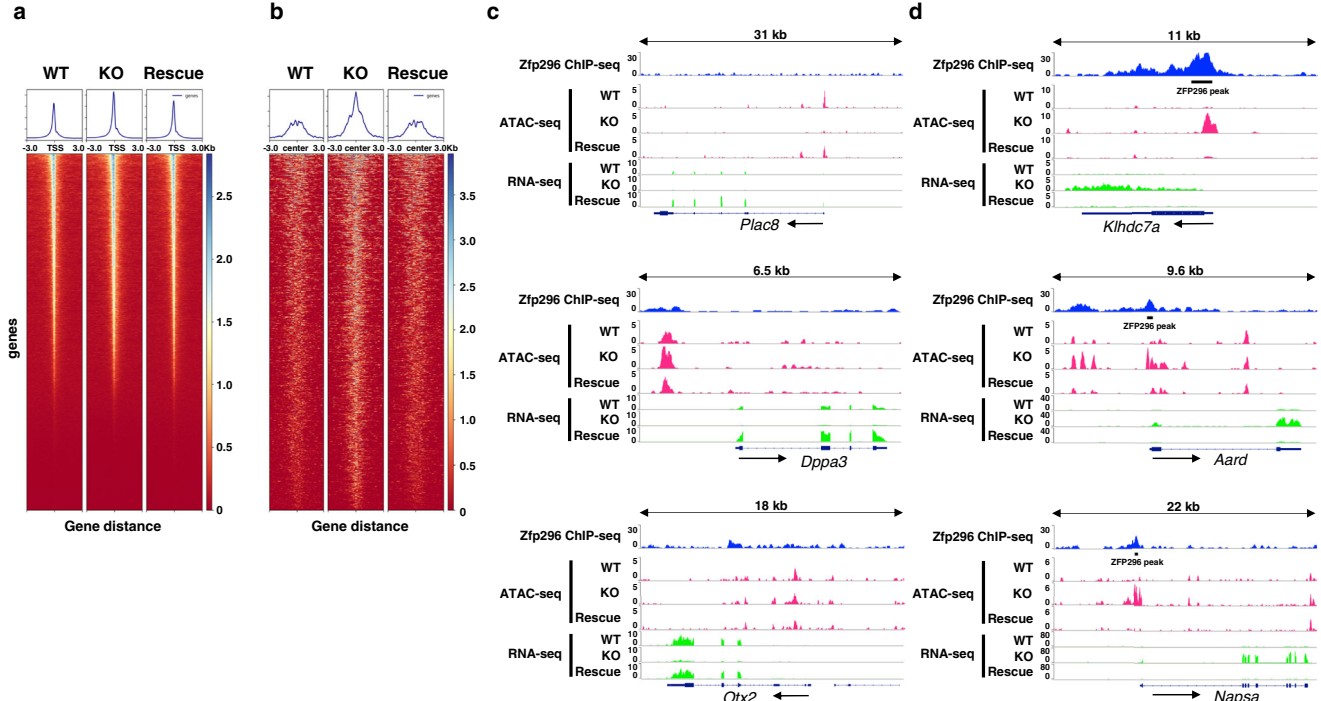

**Fig. 5 Analysis of chromatin accessibility around TSS and ZFP296 binding sites. a** Heatmaps and line plots showing average ATAC-seq signal centered on TSS. Each row corresponds to an individual gene. **b** Heatmaps and line plots showing average ATAC-seq signal centered on ZFP296 binding peaks located within 3 kb from TSS. Each row corresponds to an individual binding site. Heatmaps and line plots were made using deepTools. **c, d** Genome browser tracks of counts per million (CPM) mapped reads of ZFP296 ChIP-seq and ATAC- and RNA-seq (WT, KO #98, and Rescue #22) across three genes each that showed a marked decrease and increase in expression after *Zfp296* KO.

that do not overlap with ZFP296 binding peaks (Fig. 6a). Approximately half of the ATAC-seq peaks were commonly found in WT, KO, and Rescue ESCs. A considerable number of peaks specific to KO, Rescue, or WT ESCs were found. For the 3273 ZFP296 binding peaks, approximately 40% were not accompanied by ATAC-seq peaks, and approximately 30% were shown to overlap with common ATAC-seq peaks of WT, KO, and Rescue ESCs (Fig. 6b). Notably, 629 peaks emerged in KO cells, and 185 peaks found only in KO and Rescue cells were also considered to be induced in KO cells but retained in Rescue cells. These results indicate that a substantial fraction of the chromatin regions bound by ZFP296 become accessible after removal of ZFP296.

To further explore the relation between *Zfp296* ablation and the changes in chromatin accessibility, we compared normalized read counts in each merged ATAC-seq peak region between KO – WT, KO – Rescue, and Rescue – WT ESCs (Fig. 6c, d, left, middle, and right panels, respectively). First, we analyzed those peaks that do not overlap with ZFP296 binding peaks (Fig. 6c). For these peaks, we could not observe significant tendency of accessibility changes among WT, KO, and Rescue ESCs. However, elevated expression of a gene was mostly accompanied by increased chromatin accessibility around it in any of these ESCs, as shown by red and blue dots in Fig. 6c.

We next examined those ATAC-seq peaks that overlap with ZFP296 binding peaks and observed a considerable increase in chromatin accessibility in KO ESCs compared with WT and Rescue ESCs (Fig. 6d, left and middle panels, respectively). Furthermore, the overlapping ATAC-seq peaks located near the genes markedly upregulated in KO ESCs, such as *Klhdc7a*, *Napsa*, and *Prtg*, tended to become more accessible in KO ESCs than in WT and Rescue ESCs (Fig. 6d, red dots). Another thing we noticed for the overlapping peaks is that chromatin accessibility

was maintained at certain levels even in the presence of ZFP296. These results indicate that ZFP296 binding to TSS regions commonly attenuates chromatin accessibility, that *Zfp296* ablation converts these regions highly accessible and often enhances transcription, and that re-expression of *Zfp296* in KO cells almost completely reverts chromatin accessibility and transcription (Fig. 6d).

**Luciferase assay to evaluate the effects of ZFP296 binding on the promoter activity**. The above analysis suggested that ZFP296 functions as a negative regulator by binding to the 5'-upstream or TSS region of target genes. To pursue this possibility, we chose the *Aard* gene which was markedly (8.2-fold, KO vs. Rescue ESCs) repressed in our RNA-seq analysis. Its promoter and 5'-upstream sequences to which ZFP296 binds were cloned, and the promoter activity of this region was examined by luciferase assay in KO #98 ESCs co-expressing Flag-ZFP296 or its deletion mutants (ΔZF2-3, ΔZF4-6, or ΔZF6)[15]. The 5'-upstream region of the *Dppa3* gene was used as a control. As shown in Supplementary Fig. 5, the activity of the *Aard* promoter was markedly downregulated by ZFP296, while the activity of the *Dppa3* promoter was not affected, consistent with the result of the ChIP-seq analysis. The repressor activity of ZFP296 was almost lost by ΔZF4-6 deletion but not by ΔZF2-3 or ΔZF6 deletion, suggesting the importance of the 4th and/or 5th ZF domains in the repressor activity of ZFP296. These results demonstrated the function of ZFP296 as a repressor of transcription.

**Negative regulation of H3K9me2 by ZFP296 through interaction with histone methyltransferases**. We previously reported that ZFP296 targets heterochromatin and represses SUV39H1-dependent H3K9me2 and me3 in cultured cells[15]. We thus compared the H3K9 methylation levels among WT, KO #98, and

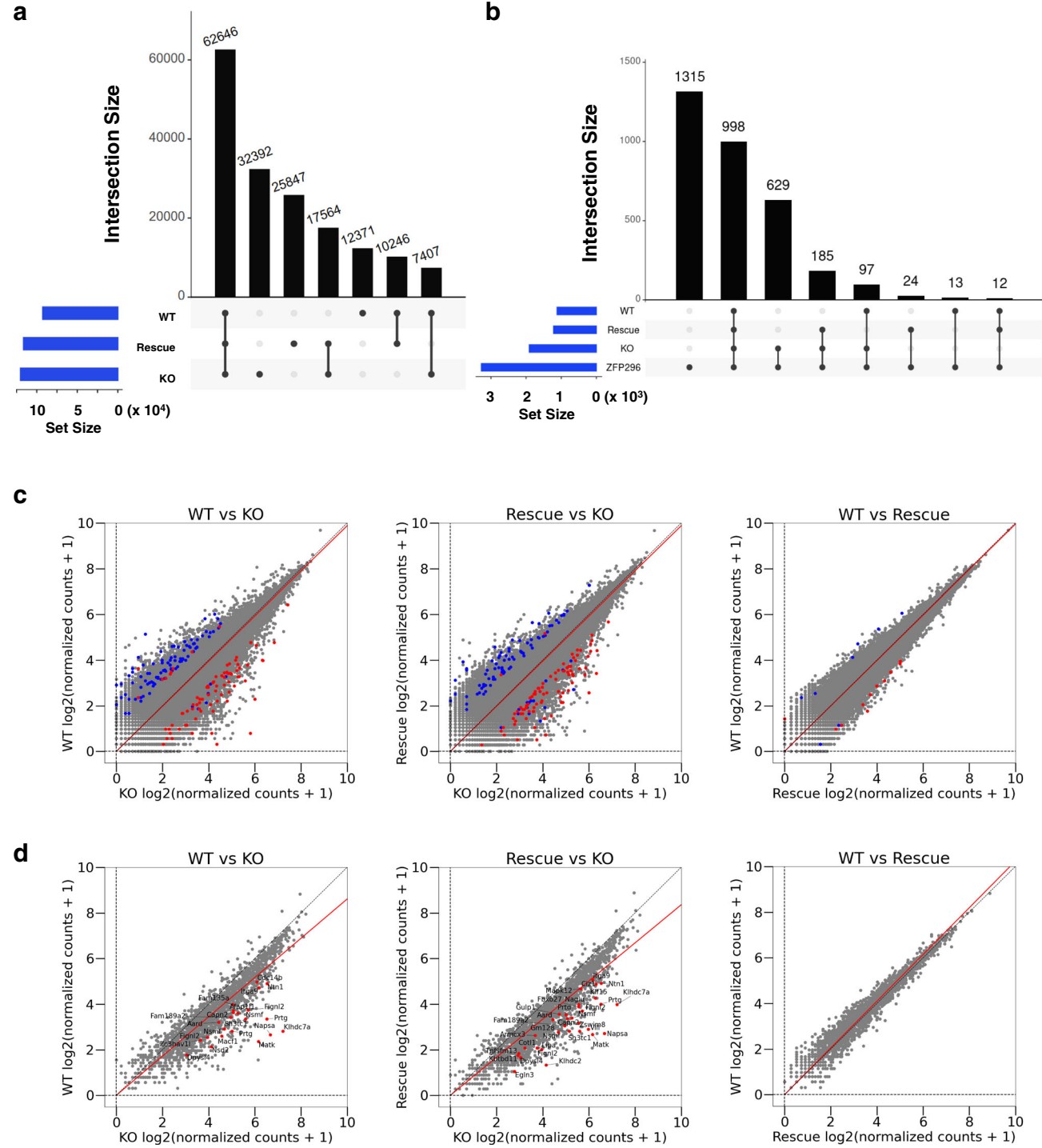

**Fig. 6 Effects of ZFP296 binding on the chromatin accessibility. a, b** Upset plots showing overlap of ATAC-seq peaks among WT, KO #98, and Rescue #22 ESCs. The ATAC-seq peaks non-overlapping and overlapping with ZFP296 binding peaks were analyzed separately. **c, d** Scatter plots showing comparison of ATAC-seq normalized read counts within each merged ATAC-seq peak between KO–WT (left panels), KO–Rescue (middle panels), and Rescue-WT (right panels). The merged peaks non-overlapping (**c**) and overlapping (**d**) with ZFP296 binding peaks were analyzed separately. Peaks near the genes that showed significant difference in expression ($P < 0.05$) were marked as follows: Red dots represent the peaks near the genes whose normalized read counts of RNA-seq were no less than 6 and more than 3-fold higher in ESCs on the X-axis than on the Y-axis for each panel; Blue dots represent the peaks near the genes whose normalized read counts of RNA-seq were no less than 6 and more than 3-fold higher in ESCs on the Y-axis than on the X-axis for each panel. Gene names were put on the red dots in the left and middle panels (**d**).

Rescue #22 ESCs by quantitative western blot analysis (Fig. 7a). The results showed a significant increase of H3K9me2 and me3 but not H3K9me1 or K27me3 in KO ESCs, suggesting that ZFP296 negatively regulates H3K9me2/3 levels.

Recent studies demonstrated that SUV39H1/H2 are crucial histone methyltransferases for H3K9me3 on pericentromeric heterochromatin and that GLP (EHMT1)/G9a (EHMT2) heteromeric complex is the primary methyltransferase for H3K9me1

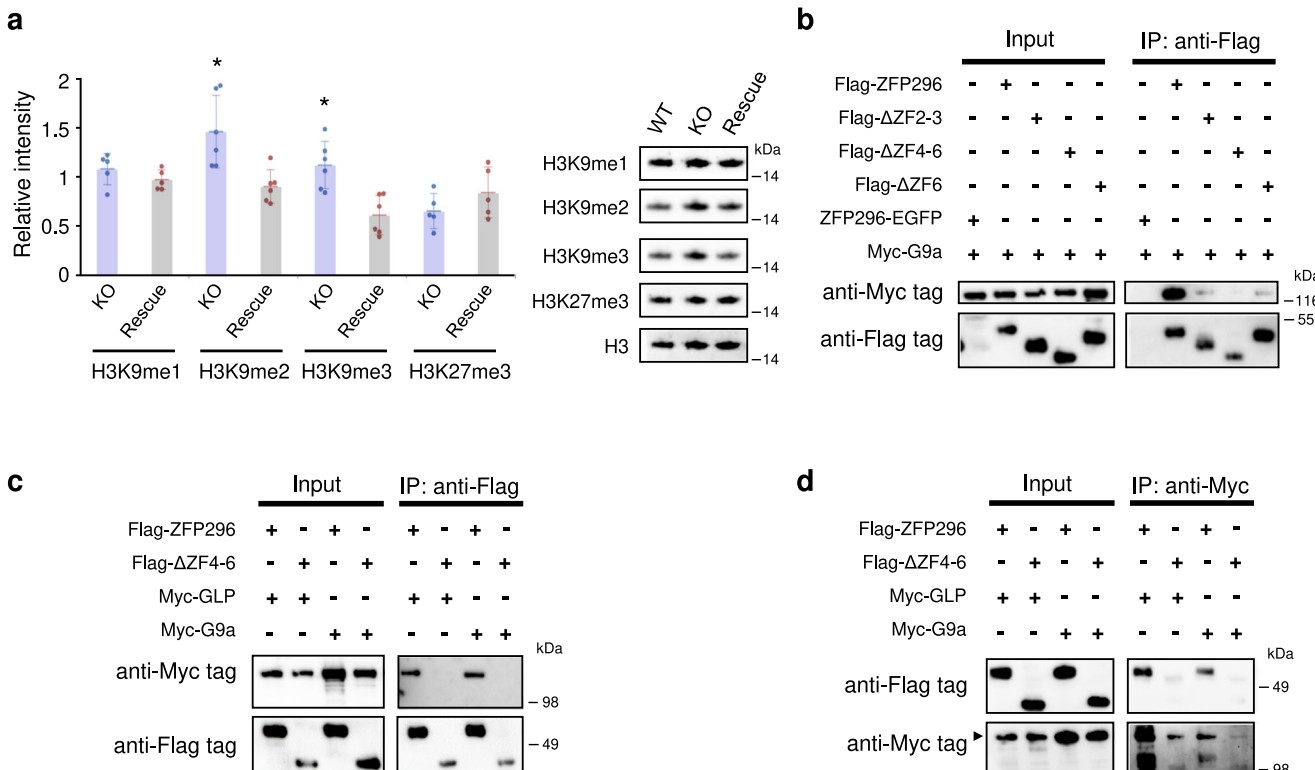

**Fig. 7 H3K9 hypermethylation in *Zfp296*-KO ESCs and binding of ZFP296 to G9a/GLP. a** Histone H3K9 methylation in the presence or absence of *Zfp296* expression. Western blot analysis was performed for H3K9me1/2/3 and H3K27me3 using nuclear extracts from WT, KO #98, and Rescue #22 ESCs. The density of each band relative to that with anti-histone H3 antibody was measured. The level of each histone modification was expressed relative to that in WT ESCs. Data represent means ± SD of five (H3K9me1 and H3K27me3) or six (H3K9me2/3) independent experiments. *P < 0.05 vs. Rescue cells by Student's *t*-test. Some of these western blot analyses are shown in the right panel. **b** Coimmunoprecipitation analysis of the interaction between ZFP296 and G9a (see Methods). Flag-ZFP296, its deletion mutants, or ZFP296-EGFP were coexpressed with Myc-G9a in 293 T cells. Nuclear extracts were immunoprecipitated with anti-Flag-tag antibody, followed by SDS-PAGE and western blotting. The membrane was treated with anti-Myc-tag antibody followed by anti-Flag-tag antibody treatment. **c, d** Coimmunoprecipitation analysis of the interaction between ZFP296 or its deletion mutant ΔZF4-6 and GLP or G9a. Nuclear extracts were immunoprecipitated with anti-Flag-tag or anti-Myc-tag antibody, followed by SDS-PAGE and western blotting. The membrane was treated with anti-Myc-tag antibody followed by anti-Flag-tag antibody treatment. Immunoprecipitation of Myc-G9a with anti-Myc-tag antibody appeared inefficient compared with that of Myc-GLP.

and H3K9me2 on euchromatin[33,34]. G9a alone or GLP alone can not exert histone methyltransferase activity, but they form a heterodimer that functions as a histone methyltransferase[35]. We presumed that ZFP296 might suppress the methyltransferase activity of G9a/GLP complex by interacting with it. To pursue this possibility, ZFP296 or its deletion mutants and G9a or GLP were coexpressed as tagged proteins in HEK293T cells, and ZFP296 was immunoprecipitated from nuclear fractions of these cells. Western blot analyses revealed that G9a and GLP were co-immunoprecipitated with ZFP296, supporting the interaction between these proteins (Fig. 7b, c, d). However, co-immunoprecipitation of G9a/GLP was considerably reduced with the ZFP296 deletion mutants, ΔZF2-3 and ΔZF6, or could not be observed with ΔZF4-6, suggesting an essential role of the 4th and/or 5th ZF domains for interaction with G9a/GLP. These results support the possibility that ZFP296 interacts with G9a and GLP and represses their methyltransferase activity.

## Discussion

In the present study, we analyzed the roles of ZFP296 in the maintenance of pluripotency of mouse ESCs. First, we found that *Zfp296* KO resulted in a dramatic change in the colony morphology of ESCs: while WT ESCs formed round compacted colonies, *Zfp296*-KO ESCs grew as flat colonies. Such abnormal colony morphology was completely reverted by exogenous

expression of *Zfp296* (Fig. 1a). It is interesting to note that the difference in the colony morphology between *Zfp296*-expressing and -deficient ESCs appears similar to that found between naïve ESCs and EpiSCs derived from postimplantation epiblast[36,37] or from ESCs in vitro[38]. Mouse EpiSCs express the core pluripotency genes *Sox2*, *Nanog*, and *Pou5f1* and can differentiate into multiple lineages in vitro. In spite of these similarities between EpiSCs and *Zfp296*-KO ESCs, they are critically different from each other in several aspects: *Zfp296*-KO ESCs like WT ESCs require LIF, but EpiSCs require activin/bFGF (basic fibroblast growth factor) for their growth and maintenance of pluripotency[39–41]; EpiSCs are able to form teratomas, while *Zfp296*-KO ESCs are not (Fig. 1d, e); *Zfp296*-KO ESCs are able to contribute to chimeras including germ cells upon blastocyst injection (Fig. 1g), but EpiSCs are unable to do so[42]. The ability of *Zfp296*-KO ESCs to contribute to chimeras may be consistent with our previous observation that *Zfp296*-deficient mice were born from *Zfp296*+/− mouse intercrosses and grew normally[15]. From these results, *Zfp296*-KO ESCs are considered to stay in a unique state of pluripotency lacking teratoma-forming ability.

The *Dppa3* gene is known as the most reliable marker for naïve ESCs[23,43]. Interestingly, our RNA-seq analysis among WT, *Zfp296*-KO, and *Zfp296*-Rescue ESCs revealed that *Zfp296* KO severely downregulated *Dppa3* expression by 12.8-fold (KO vs. WT ESCs) and that exogenous *Zfp296* expression highly elevated *Dppa3*

expression by 22.7-fold (Rescue vs. KO ESCs) (Figs. 2 and 3). Furthermore, CpG methylation analysis of the *Dppa3* promoter region demonstrated that *Zfp296* KO caused hypermethylation of this region, which was reverted by exogenous *Zfp296* expression (Fig. 3c). Such hypermethylation of the *Dppa3* promoter region has also been reported for EpiSCs. Thus, the pluripotent state of *Zfp296*-KO ESCs is very different from that of WT ESCs.

The genes downregulated upon *Zfp296* KO included the ones expressed in extraembryonic tissue (*Plac8, Fmr1nb, Krt18, Pem* (*Rhox5*), etc.), ICM (*Dppa3, Ptpn13*, etc.), and early epiblast (*Otx2, Syt13, Pou3f1, Fgf5*, etc.). Thus, many of the downregulated genes are those known to be expressed in early-stage embryos. We also noted that *Zfp296* KO reduced the expression of cytoskeleton-related genes such as *Tuba3a, Krt18, Cnn2, Myl9, Tagln, Fer1l3* (*Myof*), and *Krt19*, whose products might affect the colony morphology and cell motility. On the other hand, the genes upregulated upon *Zfp296* KO did not necessarily include the ones closely related to early embryonic development. Because *Zfp296* is highly expressed in early embryos of preimplantation period[7], we speculate that the roles of *Zfp296* in vivo and in ESCs may partly be promotion of the expression of the genes that function during implantation period and repression of the genes that are to be expressed later, which may explain elevated expression of *Sox1, Gata4, Nkx6-2, Nkx6-3, Cdx1, Sox18*, etc. observed in *Zfp296*-KO ESCs (Supplementary Fig. 3a; Fig. 2a). Therefore, *Zfp296* is likely to be involved in maintaining the metastable state of ESCs.

EpiLCs are considered as a useful system to model the transition from naïve to primed pluripotency. EpiLCs more closely parallel the early-stage epiblast than do EpiSCs, although they do not self-renew[23]. Recent studies showed that OTX2, a transcription factor essential for brain development, plays an important role in maintaining the metastable state of ESCs by antagonizing naïve pluripotency and promoting commitment to differentiation[22,44]. *Otx2* overexpression in ESCs induced exit from the naïve state and transcription of EpiLC-associated genes[22]. Interestingly, our RNA-seq analysis revealed that *Zfp296* KO caused a marked reduction in expression of *Otx2* (10.6-fold, KO vs. WT ESCs; 8.4-fold, KO vs. Rescue ESCs) and other early epiblast genes (e.g., *Pou3f1, Psmb8, Fgf5*, and *Pim2*) which are highly expressed in EpiLCs. Therefore, one of the functional roles of ZFP296 is to promote exit from naïve pluripotent state to primed one. *Zfp296*-KO ESCs expressing only low levels of *Otx2* may be stabilized in a naïve state of pluripotency. However, *Zfp296*-KO ESCs showed in vitro differentiation similar to WT ESCs (Supplementary Fig. 3). Loss-of-function mutation in *Zfp296* was shown to cause a significant impairment of in vitro development from EpiLCs to PGCLCs but not from ESCs to EpiLCs, while EpiLCs derived from *Zfp296*-null and WT ESCs display highly comparable transcriptome[16]. These results show that *Zfp296*-KO ESCs retain normal capacity of early embryonic differentiation, consistent with the chimera forming ability of *Zfp296*-KO ESCs.

Interestingly, *Gbx2* (*Gastrulation brain homeobox 2*) was found in the genes highly upregulated in *Zfp296*-KO ESCs (Fig. 2a). *Gbx2* was identified as a LIF/Stat3 downstream target. Coexpression of *Gbx2* was shown to elevate reprogramming efficiency of mouse embryonic fibroblasts to iPS cells, and forced expression of *Gbx2* was shown to reprogram EpiSCs to ground-state ESCs[45]. Therefore, upregulation of *Gbx2* seen in *Zfp296*-KO ESCs may have contributed to the maintenance of their pluripotency and germ-line chimera-forming potential.

Our ChIP-seq analysis of ZFP296 binding loci revealed that the binding of ZFP296 was preferentially seen in the TSS regions. Binding sites were frequently found in the genes upregulated but scarcely in those downregulated upon *Zfp296* KO (Fig. 2a; Fig. 5c,

d). Thus, ZFP296 could be considered a negative transcriptional regulator that binds to the promoter regions of its target genes. In fact, our luciferase assay demonstrated that the *Aard* promoter activity was repressed by ZFP296 (Supplementary Fig. 5). However, genes whose promoter regions were bound by ZFP296 did not necessarily show downregulation in *Zfp296*-expressing ESCs. It is likely that ZFP296 requires some other cofactors for its repressive activity. On the other hand, we could not find any ZFP296-binding peaks in the vicinity of some of the downregulated genes (Fig. 2a). Effects of ZFP296 on the global gene expression profiles is also likely to be mediated through indirect mechanisms.

We previously performed GST-pull down experiments using ESC nuclear extracts and recombinant GST-ZFP296 followed by liquid chromatography-tandem mass spectrometry[15]. These analyses revealed significant ZFP296 binding of various proteins including NuRD complex components such as HDAC2, CHD4, GATAD2A/B, RBBP7, MTA1/2, and MBD3, consistent with the previous report showing ZFP296 as an ESC-specific NuRD interactor[17]. By ChIP-sequencing, they showed that *Zfp296* deficiency in ESCs decreases NuRD binding both at a genome-wide scale and at ZFP296 binding sites[17]. The NuRD complex is a multi-protein transcriptional corepressor and was reported to antagonize maintenance of naïve status of ESCs[46]. Therefore, the difference in gene expression patterns between *Zfp296*-expressiong and -deficient ESCs may be mediated by the difference in genome-wide NuRD binding.

ESCs are considered to retain open and accessible chromatin configuration, which is essential to facilitate rapid activation of developmental genes upon exit from ground state pluripotency[47–49]. In our ATAC-seq analysis, chromatin accessibility around ZFP296 binding peaks was maintained at various levels in WT ESCs, but a substantial fraction of them showed dramatically increased accessibility in KO ESCs, which was often accompanied by elevated transcription of the closest genes (Fig. 6d). In Rescue ESCs, accessibility and transcription were almost completely reverted to the levels of WT cells (Figs. 2 and 6d). The *Klhdc7a, Aard*, and *Napsa* genes showed typical patterns of such coordinated changes (Fig. 5d). These results suggest that the major function of ZFP296 may be to restrict the accessibility of the TSS regions of target genes and to prevent excessive activation of these genes. Such chromatin accessibility-mediated regulation of gene expression may play an important role in maintaining the metastability of not only ESCs but also other stem cells. Although the mechanism by which ZFP296 closes the chromatin is not known, it may act in cooperation with the above-mentioned NuRD complex and/or other complexes. Elucidation of this mechanism will be an important research subject in the future.

Postulated ZFP296 binding consensus sequences were computationally obtained from the ChIP-seq analysis (Fig. 4d). Seven of the nine consensus sequences were 5 ~ 8 base long and similar in length to those obtained for other transcription factors. The other two sequences were unusually long as a consensus sequence, and we found that one of them exactly matches a part of the MSR[31]. This result agrees with our previous study which showed that ZFP296 is a component of heterochromatin[15]. Thus, ZFP296 seems to play another role as a component of constitutive heterochromatin.

In search for literature related to epigenetic regulation in ESCs, we found a close phenotypic resemblance between *Mad2l2*-KO ESCs and our *Zfp296*-KO ESCs, both of which showed the following phenotypes: (1) *Dppa3* expression was strongly suppressed but was reverted in rescue ESCs[50]; (2) H3K9me2 levels were markedly increased[50]. Furthermore, *Mad2l2*-KO mice exhibited a partial embryonic lethal phenotype and defective PGC

development, resulting in germ cell aplasia in the testes and ovaries[51–54]. As we reported previously, *Zfp296*-KO mice also showed very similar embryonic phenotypes[15]. MAD2L2 is also involved in DNA damage repair[48], but recent analysis of MAD2L2 has revealed that it works as an epigenetic regulator. MAD2L2 was shown to bind to H3K9 methyltransferases GLP and G9a and inhibit their activity, based on co-immunoprecipitation study using *Mad2l2*-transfected fibroblasts[54]. Derepression of *Dppa3* by MAD2L2 was explained by the antagonizing effect of MAD2L2 on H3K9 dimethylation via binding to GLP/G9a[50]. DPPA3 interferes with the maintenance of DNA methylation patterns by DNMT1[26]. Thus, the presence of MAD2L2 in ESCs leads to generally low levels of both DNA and H3K9 methylation[47,48]. In our *Zfp296*-KO ESCs, the global H3K9me2 levels were significantly increased (Fig. 7a), *Dppa3* expression was severely reduced, and the promoter region of *Dppa3* was de novo DNA methylated (Fig. 3c). Our co-immunoprecipitation analysis indicated interaction between ZFP296 and GLP/G9a. Therefore, the molecular mechanism for *Dppa3* regulation by ZFP296 may be analogous to that observed for MAD2L2.

Recent studies have shown that ZF family proteins, such as ZFP206[55,56], ZFP42[57], ZSCAN4 (Zinc finger and SCAN domain containing 4)[58], ZFP706[59], ZFP322a[60], ZIC2 (Zinc finger protein of the cerebellum 2)[61], ZFP281[62], etc., play crucial roles in regulating the pluripotency of ESCs. The present study revealed that ZFP296 plays important roles in maintaining the metastable pluripotent state and also in the early differentiation of ESCs via epigenetic and transcriptional regulation and that *Zfp296* KO may serve as a useful method to induce a stable state of pluripotency in ESCs without affecting their differentiation capacity.

## Methods

**Ethics statement**. Experiments involving mice were performed in accordance with relevant regulations and institutional guidelines under protocols (No. 21-089 and No. 26–066) reviewed and approved by the Institutional Animal Care and Use Committee of Osaka University.

**ES cell culture and differentiation**. The murine ESC line derived from E14tg2a, EB3[63], was cultured without feeder cells in Glasgow's Minimum Essential Medium (GMEM) (Sigma-Aldrich, St. Louis, MO) supplemented with 10% fetal bovine serum (FBS), leukemia inhibitory factor (LIF) (FUJIFILM Wako Pure Chemical, Osaka, Japan), 1% non-essential amino acids (Life Technologies, Carlsbad, CA), $100\,\mu M$ 2-mercaptoethanol, and 1 mM sodium pyruvate (Life Technologies) on gelatin-coated dishes[64]. To induce differentiation, $6.0 \times 10^4$ cells were plated into a well of a 6-well plate and electroporated into two of the *Zfp296*[+/−] ESC without LIF. On day 5 and 8, RNA extraction were performed from these cells. For embryoid body (EB) formation, approximately $3.0 \times 10^6$ trypsinized cells were seeded into a 6-cmϕ bacteriological grade dish in tetraplicate, and the culture medium without LIF was changed every 2 or 3 days starting on day 2. The numbers of beating and nonbeating EBs were counted on every 2 ~ 4 days[4].

**Targeted disruption of the *Zfp296* gene in ESCs**. One allele of the *Zfp296* gene was disrupted by inserting an IRES-βgeo-polyA cassette into exon 3 of the *Zfp296* gene as described previously[15]. To disrupt the other allele of the *Zfp296* gene, another targeting vector was constructed by replacing the IRES-βgeo-polyA cassette of the original targeting vector with the PGK-puro-polyA cassette. This second targeting vector was linearized and electroporated into two of the *Zfp296*[+/−] ESC clones. Clones resistant to $1.5\,\mu g/ml$ puromycin (Life Technologies) were screened for disruption of the WT allele of the *Zfp296* gene by long PCR. The resulting *Zfp296*[−/−] clones (KO #98 and #156) derived from different *Zfp296*[+/−] clones were further confirmed for the loss of heterozygosity and used for this study. The primers used in this experiment are listed in Supplementary Table 1.

**Generation of *Zfp296*[−/−](CAG-Zfp296) ESCs**. An expression plasmid vector for *Zfp296* was constructed using ploxP-CAG-IZ[4,65]. KO #98 and #156 ESCs were transfected with the linearized vector by electroporation and selected in the presence of 15 to $20\,\mu g/ml$ zeocin (Life Technologies) for 9 days. The resistant *Zfp296*[−/−](CAG-Zfp296) clones, Rescue #22 from KO #98 and Rescue #10 from KO #156, were expanded and used. KO #98 ESCs were also stably transfected with an EGFP expression plasmid, pCAG-EGFP-IZ. The resulting EGFP-expressing clone, KO-EGFP #26, was used.

**Teratoma formation**. For the teratoma-formation assay, $1.0 \times 10^6$ ESCs in $75\,\mu l$ phosphate-buffered saline (PBS) were injected subcutaneously into immunodeficient nude mice or histocompatible (C57BL/6 J x 129/Ola) F1 mice. Four weeks after the injection of ESCs, mice were sacrificed. The resulting tumors were removed, weighed, fixed in 20% formaldehyde overnight, and embedded in paraffin. Five-μm thick sections were cut, deparaffinized, stained with hematoxylin-eosin, and observed under a microscope.

**Generation of anti-ZFP296 antibody**. Polyclonal antibody against ZFP296 was raised by immunizing rabbits with a part of the ZFP296 protein (aa 9-82) fused to glutathione S-transferase (GST) that was produced in *E. coli* BL21. This portion of ZFP296 protein does not include any ZF domains. The resulting antiserum was immunoaffinity-purified and used for western blotting.

**Western blot analysis**. ESCs were lysed with RIPA lysis buffer (0.1% SDS, 0.5% sodium deoxycholate, 1% IGEPAL CA-630 (NP-40 substitute; MP Biomedicals, Solon, OH), 150 mM NaCl, and 50 mM Tris-HCl (pH 8.0)). Histone proteins were isolated by acid extraction protocol. Protein fractions were treated at 99 °C for 5 min in SDS sample buffer containing 80 mM DTT, subjected to SDS-PAGE, and blotted onto a polyvinylidene fluoride (PVDF) membrane (Merck Millipore, Burlington, MA). The primary and secondary antibodies used in this study were listed in Supplementary Table 2. The blots were developed using the ECL Prime Western Blotting Detection Kit (GE Healthcare, Chicago, IL) or the Western BLoT Ultra Sensitive HRP Substrate (Takara, Otsu, Japan) and visualized using X-ray films or the ChemiDoc XRS system (BioRad, Hercules, CA). Signal intensity of each band on the blots was measured and quantified with Image Lab Software (BioRad).

**Quantitative RT-PCR analysis**. Real-time PCR was performed in triplicate on the StepOnePlus Real-Time PCR System (Applied Biosystems, Foster, CA), using the FastStart Universal SYBR Green Master (Roche, Basel, Switzerland), with an initial step of 10 s at 95 °C followed by 40 cycles of 5 s at 95 °C and 30 sec at 60 °C. The expression level of each target gene was normalized to that of *Actb* (β-actin). Statistical analysis was performed by Student's *t*-test. The primers used in this study are listed in Supplementary Table 1.

**Cell proliferation assays**. ESCs were seeded in 96-well plates and incubated. After 24 or 72 h, WST-8 reagent (Cell Counting Kit; Dojindo, Kumamoto, Japan) was added to the wells. After 4 h incubation, the optical density of each well at 450 nm was measured using a microplate reader (BioRad).

**Apoptosis study**. ESCs were subjected to cleaved caspase 3 (CCP3) staining (Cell Signaling, Danvers, MA) and counterstained with DAPI (Molecular Probes, Eugene, OR). The percentage of CCP3-positive nuclei among DAPI-positive nuclei was monitored and quantified using Image J software.

**RNA-sequencing and differential expression analysis**. Total RNA was extracted from WT, KO #98, and Rescue #22 ESCs using an RNA purification kit (NucleoSpin RNA Plus; MACHEREY-NAGEL, Düren, Germany). The quality of the purified RNA was confirmed using an Agilent 2100 Bioanalyzer (Agilent, Santa Clara, CA). Libraries for sequencing were constructed from the fragmented RNA using the TruSeq Stranded mRNA Sample Prep Kit (Illumina, San Diego, CA). The resulting libraries were sequenced on a HiSeq 3000 genome analyzer (Illumina) with a 100-base single-end read setting. Similarly, RNA from KO #156 and Rescue #10 ESCs was analyzed on a NovaSeq 6000 genome analyzer (Illumina) with a 101-base single-end read setting. Sequenced reads were mapped to the mouse reference genome sequence (mm10) using TopHat ver. 2.0.13 and quantified using Subread featureCounts v2.0.3 with GENCODE vM25 gene annotation. Read counts were transformed to the number of fragments per kilobase of exon and then analyzed with edgeR v3.36.0 in R v4.1.2, employing the estimateGLMCommonDisp with method = "deviance" robust=TRUE and subset=NULL options and exact test for identification of differentially expressed genes.

**Fluorescence microscopic analysis**. Cells were cultured in 3.5 cm film-bottom culture dishes (FD10300; Matsunami Co., Osaka, Japan), fixed in 4% paraformaldehyde, and blocked in PBS containing 10% goat or donkey serum and 3% bovine serum albumin for 1 h at room temperature. The samples were incubated with primary antibodies overnight at 4 °C, washed, and incubated with Alexa Fluor-conjugated secondary antibodies for 1 h at room temperature. Cells were counterstained with $1\,\mu g/ml$ DAPI for 15 min at room temperature prior to mounting. All imaging was performed using a BZ-9000 multifunctional microscope (Keyence, Osaka, Japan) or an FV1000 confocal microscope (Olympus, Tokyo, Japan).

**Bisulfite sequencing**. Bisulfite treatment of the genomic DNA isolated from ESCs was carried out using the EpiTect Bisulfite Kit (Qiagen, Hilden, Germany) according to the manufacturer's instructions. The CpG islands in the upstream

region of the *Dppa3* gene were analyzed[25]. The primers used to amplify this region were 5'-TTGGTTTTAGAATTTAGAGATTTATT-3' and 5'-CCAATTAACAATC AATCTATAAATA-3'. The PCR products were cloned into pBlueScript and sequenced. The extent of methylation was assessed by C to T conversion at the CpG sites in the amplified region.

**Chromatin immunoprecipitation-sequencing (ChIP-seq)**. We prepared a plasmid pCAG-Ty1-Zfp296-IZ expressing ZFP296 connected to Ty1 tag at the N-terminal. This plasmid was stably transfected into KO #98 ESCs, resulting in *Zfp296*$^{-/-}$(CAG-Ty1-Zfp296) ESCs. DNA samples for ChIP-seq were prepared from these cells using the SimpleChIP Enzymatic Chromatin IP Kit (Cat#9003; Cell Signaling) according to the manufacturer's instructions. Briefly, cells were treated with formaldehyde and lysed. Nuclei were treated with micrococcal nuclease and then sonicated for 15 s x 50% pulse x 15 times using an ultrasonic homogenizer (Sonifier 250; Branson Ultrasonics, Brookfield, CT). After centrifugation, the supernatant was incubated with anti-Ty1-tag monoclonal antibody (Diagenode, Denville, NJ) overnight. Antibody-bound chromatin fragments were collected using Protein G-conjugated magnetic beads (Dynabeads Protein G; Life Technologies), and DNA was extracted from them. For ChIP-seq, libraries were prepared using the TruSeq ChIP Sample Prep Kit (Illumina) and sequenced using the Illumina HiSeq 4000 sequencer as 101-base single-end reads. Sequence reads were mapped to the mm10 mouse genome using Stampy with default parameters[66]. Only reads that uniquely mapped to the mouse genome were used. Peak calling was performed using MACS2 software with default parameters to call areas of enrichment relative to the genome background. Peaks with $q$-value < 0.1 were identified. The genomic distribution of ChIP-seq peaks compared to gene annotations was determined using Cis-regulatory Element Annotation System (CEAS). Consensus DNA binding motifs for ZFP296 were identified using MEME[29] and DREME[30] in the retrieved sequences of the ZFP296 binding regions.

**Analysis of 5mC and 5hmC levels by dot blot assay**. Control 5mC and 5hmC DNA (Zymo Research, Irvine, CA) and genomic DNA from WT, KO #98 and #156, and Rescue #22 and #10 ESCs were denatured with 0.1 N NaOH. Two-fold serially diluted DNAs were spotted onto positively charged nylon membranes (Roche). The membranes were baked at 80 °C and blocked with Blocking One (Nacalai Tesque, Kyoto, Japan) for 30 min at room temperature. The membranes were then incubated with mouse monoclonal antibody against 5mC (Merck Millipore) or rabbit polyclonal antibody against 5hmC (Active Motif, Carlsbad, CA) overnight at 4 °C. After washing 3 times with Tris-buffered saline containing 0.05% Tween 20 (TTBS), membranes were incubated with HRP-conjugated anti-mouse or anti-rabbit IgG antibody, respectively. After washing 3 times with TTBS, the membranes were developed using the ECL Western Blotting Detection Kit (GE Healthcare), and the dots were visualized by exposure to X-ray films.

**ATAC-sequencing**. ATAC-sequencing was applied to WT, KO #98, and Rescue #22 ESCs using the ATAC-Seq Kit (Cat#53150; Active Motif) according to the manufacturer's instructions. In brief, nuclei were isolated from $1 \times 10^5$ cells and subjected to tagmentation reaction at 37 °C for 40 min using tagmentase-loaded (Diagenode). DNA was purified from the reaction mixture using DNA purification columns, and transposed fragments were amplified by 10 cycles of PCR using indexed primer pairs. The resulting libraries were sequenced using the Illumina NovaSeq 6000 platform in a 151-base paired-end mode. Adapter sequences were removed from raw reads using Cutadapt v3.2, specifying Nextera adapter sequences and a minimum length of 30 bases. Trimmed reads were mapped to the mouse mm10 genome using Bowtie2 v2.3.5.1 with --no-discordant --no-mixed --dovetail and --very-sensitive options. PCR duplicated reads were removed using Picard tools v1.128 with the MarkDuplicates function, and then mitochondrial reads were removed. Reads with fragment length below 100 bp were extracted as nucleosome-free reads. The extracted mapped reads were shifted using deepTools v3.4.3 alignmentSieve to adjust for transposase binding offset. Peaks were identified using MACS2 in narrow mode, and peak files across the samples were merged using Bedtools v2.25.0 to create a consensus peak file. Consensus peaks were annotated using HOMER, annotatePeaks.pl. Enrichment heatmaps and average plots were generated in deepTools for the consensus peaks as well as the peaks that overlap with ChIP-seq peaks. Reads within consensus peaks were quantified using Subread featureCounts v2.0.3. Read counts were analyzed with edgeR v3.36.0 in R v4.1.2, employing the estimateGLMCommonDisp with method = "deviance" robust=TRUE and subset=NULL options and exact test for identification of differentially accessible regions.

**Luciferase assay**. The promoter region of the mouse *Aard* gene (− 2933 to −37 bp upstream of the start codon) was obtained from the C57BL6/J mouse genomic DNA by PCR and inserted into the multi-cloning sites of the pGL3-Basic vector (Cat#E1751; Promega, Madison, WI). Similarly, the 5'-upstream region of the mouse *Dppa3* gene (− 4324 to −53 bp upstream of the TATA box) was obtained and inserted into the upstream of the minimal promoter of the pGL3-promoter vector (Cat#E1761; Promega). These promoter and 5'-upstream regions obtained by PCR-amplification were confirmed by sequencing. The primers used in this experiment are listed in Supplementary Table 1. The expression plasmid vectors used for the reporter assays were as follows: pCAG-Flag-Zfp296, pCAG-

Flag-Zfp296ΔZinc6, pCAG-Flag-Zfp296ΔZinc4-6, pCAG-Flag-Zfp296ΔZinc2-3[15], and pCAG-Zfp296. KO #98 ESCs ($1 \times 10^5$ cells) were plated on 24-well plates with transfection mixture containing 500 ng of a reporter vector, 40 ng of a *Zfp296* expression vector, 400 ng of pBlueScript, 50 ng of pRL-TK normalization vector, and 7.2 μl of Polyethylenimine Max (PEI-Max 1 mg/ml; Polysciences, Inc., Warrington, PA) per well. After 48 h incubation, the total cell extracts were subjected to luciferase assays using the Dual-Luciferase Reporter Assay System (Promega).

**Co-immunoprecipitation assay**. G9a and GLP cDNAs were amplified by RT-PCR and cloned into the pCAG-Myc-IRES-puro vector[67]. HEK293T cells were maintained in Dulbecco's Modified Eagle Medium (DMEM) high glucose (Sigma-Aldrich) supplemented with 10% FBS. Cells were grown to 80% confluency and transfected with pCAG-Flag-Zfp296, pCAG-Flag-Zfp296ΔZinc2-3, pCAG-Flag-Zfp296ΔZinc4-6, or pCAG-Flag-Zfp296ΔZinc6[15] in combination with pCAG-Myc-G9a or pCAG-Myc-GLP using PEI-Max following the manufacturer's instructions. On the next day, culture medium was removed and replenished. Two days later, the cells were harvested with a cell scraper, washed with PBS, and suspended in hypotonic buffer containing 10 mM HEPES buffer (pH 7.9), 10 mM KCl, 1.5 mM MgCl$_2$, and 1 mM DTT supplemented with protease inhibitor cocktail (Merck, Darmstadt, Germany) and 1 mM phenylmethylsulfonyl fluoride (PMSF) (Merck). Cells were lysed by adding 0.1% IGEPAL CA-630 and vortexing briefly. The lysates were centrifuged at 10,000 x $g$ for 30 s at 4 °C. The pellet containing nuclei was washed once with hypotonic buffer and suspended in high-salt buffer containing 20 mM Tris-HCl buffer (pH 8.1), 500 mM NaCl, 3 mM MgCl$_2$, 0.5% IGEPAL CA-630, and 1 mM DTT supplemented with 100 U/ml Cryonase (cold-active nuclease; Takara), protease inhibitor cocktail, and 1 mM PMSF. The nuclei were incubated on ice for 20 min with occasional vortexing. After centrifugation, the supernatants were diluted 4-fold with buffer containing 20 mM Tris-HCl buffer (pH 7.9) and 0.1 mM DTT supplemented with protease inhibitor cocktail and were incubated with magnetic beads conjugated with anti-DYKDDDDK (Flag) antibody (clone 1E6) (FUJI-FILM Wako) or anti-Myc-tag antibody (MBL, Nagoya, Japan) for 3 h while rotating at 4 °C. The beads were washed three times with wash buffer (25 mM Tris-HCl (pH 7.4), 2.68 mM KCl, and 137 mM NaCl), suspended in SDS sample buffer containing 80 mM DTT, and heated at 99 °C for 5 min. Beads were magnetically separated, and the supernatants were subjected to western blot analysis as described above.

**Statistics and reproducibility**. Experimental data were reported as means ± SD with the number of replicates provided in the figure legends. Statistical differences were determined by Student's $t$-test or Tukey's test. A value of $P < 0.05$ was considered significant.

**Reporting summary**. Further information on research design is available in the Nature Portfolio Reporting Summary linked to this article.

## Data availability
The numerical source data for the graphs presented in this study can be found in Supplementary Data. Supplementary Fig. 6 contains uncropped western blot images with size markers. Sequencing data of RNA-seq, ChIP-seq, and ATAC-seq generated in this study have been deposited in the NCBI GEO repository under the accession number GSE231412. All other data supporting the findings of this study are available from the corresponding author upon reasonable request.

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

## Acknowledgements

We are grateful to Mr. Masafumi Ashida for technical assistance. We acknowledge the NGS core facility of the Genome Information Research Center at the Research Institute for Microbial Diseases of Osaka University for the support in RNA-seq and ATAC-seq. High-throughput DNA sequencing for ChIP-seq was conducted at Hokkaido System Science Co., Ltd. (Sapporo, Japan). This research was supported in part by Grants-in-Aid for Scientific Research from the Japan Society for the Promotion of Science (No. 25460364) to J.M.

## Author contributions

S.M., F.T., T.Ma., T.Mi, and J.M. performed the experiments and analyzed the data. H.Y. and D.M. analyzed the data of RNA-seq, ChIP-seq, and ATAC-seq. S.M. and J.M. generated figures and wrote the manuscript. J.M. conceived the study and supervised the work.

## Competing interests

The authors declare no competing interests.
