## [Peer Review File · Communications Biology]

Reviewers' comments:

Reviewer #1 (Remarks to the Author):

Miyazaki et al. explored the role of Zfp296 in mouse ESCs. Their findings include: Zfp296 KO ESCs grew as flat colonies, which could be reversed by exogenous expression of Zfp296; Zfp296 KO ESCs could not form teratomas, but efficiently produced germline-competent chimeric mice; Zfp296 down-regulates the expression of Otx2, Dppa3, Pou3f1, Fgf5, and Dnmt3b in ESCs; ZFP296 binds the vicinity of the transcription start sites of a number of genes, probably acting as a transcriptional repressor; ZFP296 interacts with G9a and GLP. Some of these findings might be interesting, however the current version of manuscript is preliminary for the publication in *Communications Biology*. The following are several comments:

1. Due to the heterogeneous phenotypes of Zfp296 KO ESCs, RNA-sequencing for KO #156 and Rescue #10 ESCs is suggested.
2. A systematical analysis of RNA-seq, Chip-seq data and the specific targets should be significantly improved the quality of the manuscript.
3. The expression levels of Dppa3 (Figure 2) is not consistent with DNA methylation levels of the Dppa3 promoter region (Figure 3c) in these cells. These results should be discussed.
4. Figure 3b, the loading control should be provided. The quantification of these data are suggested.

Reviewer #2 (Remarks to the Author):

Miyazaki S et al. studied the role of the zinc finger-type protein Zfp296 in ESCs, in which this factor is highly expressed. They established Zfp296-knockout (KO) ESCs and studied the consequences at the cellular level (in vitro and in vivo), and the molecular level (transcriptomics and genomics). The authors claim that Zfp296 KO induced phenotypic change of ESCs. Zfp296-KO cells proliferate more, show less apoptosis, grow as flat instead of round colonies and tend to form less teratoma in vivo. When introduced into blastocysts, Zfp296-KO cells participate efficiently in the generation of germline-competent chimeric mice. RNAseq and RT-qPCR assays showed that Zfp296-KO induced downregulation of many early epiblast-marker genes and upregulation of genes that are normally repressed in epiblast-like cells (EpiLCs). The authors have then studied in more detail the Dppa3 gene which is involved in global DNA demethylation in ESCs and strongly repressed upon Zfp296 KO. ChIP assays showed that Zfp296 is predominantly localized close to TSS and suggested that Zfp296 acts as a transcriptional repressor. Surprisingly, Zfp296-KO induced a global increase in the repressive histone marks H3K9me2 and H3K9me3. Furthermore, co-IP assays showed that ZFP296 interacts with the KMTs G9a and GLP, which are responsible of H3K9. The authors concluded that ZFP296 is a transcriptional repressor that is essential in maintaining open chromatin states in ESCs by repressing H3K9 methylation, including at the Dppa3 gene. The Zfp296-KO ESCs express less early epiblast genes and retain "unique ground-state pluripotency".

The findings could be of potential interest. However, in its current form the study fails to make a compelling case at present. While the cellular studies seem of good quality, the molecular ones need more controls. Indeed, many experiments are unsatisfactory, the most essential ones provide rather indirect evidences. The work needs additional key data to consolidate the main conclusion, namely the role of Zfp296 in transcriptional regulation. Moreover, many data seem mis- or over-interpreted and important points need to be clarified.

The following few points are intended as constructive suggestions to improve this work (I apologize in advance if missed some details within the provided files):

GENERAL COMMENTS:

- The proposed model concerning the molecular role of ZFP296 is counter intuitive. Indeed, the authors claim that ZFP296 acts as a transcriptional repressor and in the same time ZFP296 inhibits one of the most characterized repressive histone marks, which is H3K9 methylation (via interaction and inhibition of G9a/GLP). They claimed on lines 76-77: Our results showed that ZFP296 is a component of heterochromatin and negatively regulates H3K9 methylation. While I am really not at all against new concepts and unexpected results, I have some issues with the authors proposed model here. Indeed, how the KO of a transcriptional repressor and heterochromatin factor could induce a global increase in one of the strongest repressive histone marks H3K9me2 and H3K9me3. Thus, the authors must provide more experimental evidence support their conclusions. They must at least perform H3K9-methylation ChIP-qPCR (H3K9me2 and/or H3K9me3) on the already studied genes (DPP3 and Fig 5). They could also perform H3K9me2 and/or H3K9me3 ChIPseq and cross the data with their RNAseq data. Without H3K9me ChIP-qPCR ChIPseq data the authors should reformulate their conclusions, especially in the MS abstract.
- ATAC-seq should be performed to support the authors statement that "ZFP296 is also considered to contribute to the maintenance of a generally open chromatin configuration ».
- The assertion that ZFP296 could inhibit the G9a/GLP heterodimerization must be checked in vitro using recombinant proteins. In addition, authors should cross ZFP296 and G9A/GLP ChIPseq data and compare the ZFP296 and G9a/GLP KO induced phenotypes in ESCs.
- Line 315: "The results showed a significant increase of H3K9me2 and me3, but not H3K9me1 or K27me3, in KO ESCs, suggesting that ZFP296 negatively regulates H3K9me2/3 levels": at least discuss how these data fit with the lack of any difference in 5mc levels and an increase in 5hmc in KO cells (Supplemental Fig 5).
- To gain more insights into ZFP296 direct versus indirect gene targets cross the RNAseq and ChIPseq data.

SPECIFIC COMMENTS:

- Fig 3b: "The percentage of beating EBs was considerably lower in Rescue ESCs, compared with the other cell lines during the observation period...": this result seems surprising, where the WT and KO are comparable while the rescue behaves differently for WT and KO! The authors conclusion that "... suggesting that persistent expression of Zfp296 suppresses mesodermal differentiation" is not satisfactory at all. Please comment.
- Lines 106-108: "Interestingly, Zfp296-overexpressing ESCs formed homogeneous colonies with compacted morphology": please cite a reference.
- Line 126: show the karyotype data.
- Lines 203-204 (and elsewhere): "The genes whose expression was the most severely affected are shown in Fig. 2a », please state the chosen thresholds and p-values instead of "most severely affected".
- Figure issues: Fig 1b and 1C, improve the Y-axis legend (proliferation and apoptosis (%) are not

informative enough).

- Many figures do not contain any statistical analyses of the data unless I missed them (Figures 1d, 2 a-b-c, all Supplemental Fig 3). A "Statistical analyses" chapter must be included in the Methods section.

- Fig 3a: IF quality does not meet publication standards. Show different enlargements, quantify IF signals and perform statistical analyses, comment the different DPP3 IF intensity signal within WT and rescue cells...

- Fig 3b: western blot (WB) loading control, signal quantification and the number of replicates are missing. WB signals must be quantified in at least 3 independent experiments and statistically analyzed. The number of experiments must be stated in the figure legend.

- Fig 5 a-b: confirm the RNAseq data by RT-qPCR. Improve the Y-axis scale form.

- Fig 5c and Fig 6a: Y-axis legend is missing.

- Fig 6a: show WB image examples.

- General: it is not correct to state "genes down- or up-regulated by ZFP296" when mentioning genes down or up-regulated upon ZFP296 KO. Please modify this all over the MS.

- The ChIPseq raw data deposition is not stated in the MS.

- Page 29: "Data availability : The data supporting the findings of this study are available from the corresponding author upon reasonable request": this is somehow surprising since the authors are supposed to make available all their raw data.

- The discussion section is way too speculative, especially when it is based on the RNAseq data (regarding Otx2 and Gbx2 gene expression for example) without any confirmation. It should be reduced.

MINOR COMMENTS:

- Fig 4e legend title "ZFP296 binding to heterochromatin" is not correct. An IF does not show binding but localization. Please modify.

- Supplemental Table 1: add a legend.

- Line 201: state which phenotypes.

- Write Zfp296 instead of ZFP296 when mentioning the protein.

- General: please show the color code meaning in the heatmap figures (2a, Supplemental 5b).

Reviewer #3 (Remarks to the Author):

This manuscript describes the characterization of Zfp296 knockout embryonic stem cells (ESC). The present manuscript is the continuation of studies previously performed by the same research group aimed at understanding the role of Zfp296 during development and in ESCs.

The authors have studied the role of Zfp296 in maintenance of self-renewal and pluripotency. They have generated Zfp296 KO ESCs and show, that these cells are able to retain a pluripotent state and germline competence. Furthermore, Zfp296 deficiency induces the downregulation of early epiblast-marker genes. Zfp296 acts as a transcriptional repressor, interacts with G9a and GLP, and plays a role in maintaining open chromatin structure in ESCs. By using deletion mutants, the authors show the functional importance of the 4th and/or 5th ZF domains of ZFP296.

The work presented is of very good quality and the data is interesting. Nevertheless, I am not fully convinced with the conclusions presented.

The authors state in the title that "Zfp296-KO ESCs retain unique ground-state", my impression is that absence of Zfp296 impairs differentiation rather than contributing to pluripotency. This would for example explain why the KO cells are unable to undergo teratoma formation. I would be therefore careful with the title.

The changes in morphology are interesting and indeed reflect the differentiation status of ESC populations, but are not conclusive. Zfp296-overexpressing ESCs as well as rescued Zfp296^{-/-} cells form homogeneous colonies similar to the WT ESCs (Fig.1). Both WT and knockout ESCs are germline competent independent of their morphology, therefore I doubt that Zfp296 has a role in maintaining pluripotency even if the morphology of the colonies is changing. A staining of the cells with markers like Oct-4, SSEA1, Rex1 and Nanog of the colonies could give some important additional information.

The fact that Zfp296 KO ESCs are not able to form teratoma is interesting, but at least for mouse ESCs the proof of pluripotency is the capacity to form germline competent chimera, and the authors show that the absence of Zfp296 does not impair this capacity. Also here, the interpretation could be that Zfp296 is not necessary for keeping a pluripotent state.

In this context, how does the stable expression of Zfp296 (expression that cannot be downregulated upon injection in the blastocyst, e.g. rescue lines) influence pre-/post-implantation embryonic development? Or, at least are Zfp296 KO cells able to differentiate in vitro?

Is the expression of Zfp296 in LIF/serum similar to 2i/LIF (ground state)?

Open chromatin is a characteristic of pluripotent cells but it is not a determinant of pluripotency. The data presented highlighting the role of Zfp296 in maintaining an open chromatin structure are well described. But if this is the reason why Zfp296 KO ESCs retain unique ground-state pluripotency cannot be determined by the experiments presented in the manuscript.

How is the methylation changing upon differentiation of Zfp296^{-/-} cells in vitro? Are the cells able to terminally differentiate? Can they be upon differentiation brought back to a pluripotent state upon addition of serum/LIF?

A more general comment that should be addressed in the discussion: It is not clear to me how is it possible that Zfp296 knockout ESCs are able to generate germline competent chimeric animals when Zfp296 was previously shown to be responsible for acquisition of PGC fate and its abrogation leading to failure in activating germline genes and inducing loss of germ cell identity. Are here compensatory effects playing a role?

In conclusion, the data presented is very interesting but not conclusive. Despite the changes in morphology, the inability to form teratoma and the increase in histone H3K9 di- and trimethylation Zfp296 KO cells retain a pluripotent state and generate germline competent chimera. The data can be interpreted also in the direction that Zfp296 is expressed in ESCs and its depletion does not impair their pluripotent state, means Zfp296 is not important for self-renewal and pluripotency. I therefore would strongly recommend the authors not to use the concept of retaining "unique ground-state" pluripotency in the context of Zfp296 KO cells but rather state that Zfp296 cells are pluripotent.

Responses to the Reviewers' Comments

We are very grateful to the reviewers for their constructive comments. In response to these comments, we performed several additional experiments: i) RNA-sequence of another *Zfp296*-KO clone (#156) and its rescue clone (#10); ii) Real-time PCR analysis of RNA from wild-type (WT), KO #98, and Rescue #22 ESCs and WT, KO #156, and Rescue #10 ESCs; iii) ATAC-sequence of WT, KO, and Rescue ESCs; iv) Immunofluorescence analysis of DPPA3 expression in WT, KO, and Rescue ESCs; v) Western blot analysis of DPPA3 and H3K9 methylation in WT, KO, and Rescue ESCs. Systematical analyses of the results of RNA-seq, ChIP-seq, and ATAC-seq have been conducted in cooperation with Drs. H. Yamano and D. Motooka who joined this study as coauthors. Based on the results obtained by these experiments, we have extensively revised our manuscript. According to the Reviewer's comment and the data obtained from ATAC-seq, we have also changed the title to "***Zfp296* knockout enhances chromatin accessibility and induces a unique state of pluripotency in embryonic stem cells.**"

We have revised most Figures and Tables and added three Figures in the revised manuscript as follows:

Fig. 2a: RNA-seq data from KO #156 and Rescue #10 were added. The most severely affected genes were listed. The listed genes are mostly the same with those listed in the previous version of Fig. 2a.

Figs. 2b: Several genes which showed elevated expression upon *Zfp296* KO were indicated.

Fig. 3a: We have replaced the image data of immunofluorescence analysis of DPPA3 expression in WT, KO, and Rescue ESCs to new ones. We believe the revised data show much better resolution of DPPA3 staining.

Fig. 3b: We performed western blot analysis of DPPA3 in which Lamin B1, a nuclear protein, served as the loading control. Quantitative analysis of the results has been added as a bar graph.

Fig. 3d: Dot blot analysis of 5mC and 5hmC was moved from supplementary figure to Fig. 3d.

Fig. 5 (new figure): a, b Analysis of chromatin accessibility around TSS and ZFP296 binding sites. **c, d** Genome browser tracks of mapped reads of ZFP296 ChIP-seq and ATAC- and RNA-seq for six genes are shown.

Fig. 6 (new figure): a, b Upset plots showing overlap of ATAC-seq peaks among WT, KO, and Rescue ESCs. **c, d** Scatter plots showing the effects of ZFP296 binding on the chromatin accessibility.

Fig. 7: Fig. 6 in the first submission was renumbered to Fig. 7. We performed additional western blot analyses for H3K9me1/2/3 and K27me3. The result shown in new Fig. 7a is almost the same with the previous one. An example of these western blot analyses is shown in the right

panel of Fig. 7a.

Supplementary Fig. 4: The original Supplementary Fig. 4 was removed. Real-time PCR analysis of 8 genes was performed for WT, KO #98 and Rescue #22 ESCs and also for WT, KO #156 and Rescue #10 ESCs. The results are shown in Supplementary Fig. 4a, b.

Instead, we removed two Figures, Fig. 5 and Supplementary Fig. 5 in the first submission. Luciferase assay in Fig. 5 was moved to Supplementary Fig. 5. Dot blot analysis of 5mC and 5hmC in Supplementary Fig. 5 was moved to Fig. 3d.

Supplementary Table 1 (RNA-seq data) was removed because the RNA-seq data have been deposited in the NCBI GEO (see below). Supplementary Table 2 (primer list) and Supplementary Table 3 (antibody list) were renumbered to Supplementary Table 1 and Supplementary Table 2, respectively.

The raw data of RNA-seq, ChIP-seq, and ATAC-seq have been deposited in the NCBI GEO site. The following reviewer token can be used to access the link below.

<https://www.ncbi.nlm.nih.gov/geo/query/acc.cgi?acc=GSE231412>

reviewer token: gpslomkydhubrgr

These data will be made available to the public once publication date is determined. The deposition information is described in the last part of the Methods section (“Data availability”).

Our responses to each comment were described below in detail.

Reviewer #1:

Comment (1): Due to the heterogeneous phenotypes of *Zfp296* KO ESCs, RNA-sequencing for KO #156 and Rescue #10 ESCs is suggested.

Response (1): According to the reviewer’s suggestion, we performed RNA-seq on KO #156 and Rescue #10 RSCs. The results of KO #156 and Rescue #10 along with WT, KO #98 and Rescue #22 are shown in revised Fig. 2a and the raw data of RNA-seq have been uploaded on the NCBI GEO site as described above. We do not think that the phenotypes of *Zfp296*-KO ESCs are considerably heterogeneous between KO #98 and KO #156.

Comment (2): A systematical analysis of RNA-seq, Chip-seq data and the specific targets should be significantly improved the quality of the manuscript.

Response (2): The systematical analyses of the results of RNA-seq, ChIP-seq, and ATAC-seq have been conducted in cooperation with Drs. H. Yamano and D. Motooka who are specialized in this field and participated in the present study as coauthors. The results are mainly shown in

revised Figs. 5 and 6.

Comment (3): The expression levels of *Dppa3* (Figure 2) is not consistent with DNA methylation levels of the *Dppa3* promoter region (Figure 3c) in these cells. These results should be discussed.

Response (3): We agree with the Reviewer's concern. The levels of *Dppa3* mRNA and protein in Rescue ESCs were comparable to those in WT ESCs (Fig. 3b; Supplementary Fig. 4). However, DNA methylation levels of the *Dppa3* promoter region were considerably higher in Rescue ESCs than in WT ESCs. We considered possible reasons which may explain this inconsistency, but it seemed difficult to explain it. Therefore, we have added the following description "Considering that the *Dppa3* expression in Rescue ESCs was comparable to that in WT ESCs, DNA methylation levels of the promoter region in Rescue ESCs seemed high. However, we do not know the exact reason for this inconsistency."

Comment (4): Figure 3b, the loading control should be provided. The quantification of these data are suggested.

Response (4): According to the Reviewer's suggestion, we carried out western blotting experiments including the loading control, Lamin B1, as shown in Fig. 3b. The band intensity was measured using the Image Lab software. The quantitative results are displayed as a bar graph ($n = 3$; Fig. 3b, lower panel).

Reviewer #2:

GENERAL COMMENTS:

Comment (1): The proposed model concerning the molecular role of ZFP296 is counter intuitive. Indeed, the authors claim that ZFP296 acts as a transcriptional repressor and in the same time ZFP296 inhibits one of the most characterized repressive histone marks, which is H3K9 methylation (via interaction and inhibition of G9a/GLP). They claimed on lines 76-77: Our results showed that ZFP296 is a component of heterochromatin and negatively regulates H3K9 methylation. While I am really not at all against new concepts and unexpected results, I have some issues with the authors proposed model here. Indeed, how the KO of a transcriptional repressor and heterochromatin factor could induce a global increase in one of the strongest repressive histone marks H3K9me2 and H3K9me3. Thus, the authors must provide more experimental evidence support their conclusions. They must at least perform H3K9-methylation ChIP-qPCR (H3K9me2 and/or H3K9me3) on the already studied genes (*DPP3* and Fig 5). They could also perform H3K9me2 and/or H3K9me3 ChIPseq and cross the data with their RNAseq data. Without H3K9me ChIP-qPCR ChIPseq data the authors should reformulate their

conclusions, especially in the MS abstract.

Response (1): Since there is no significant binding site for ZFP296 in the promoter or upstream region of *Dppa3* (Fig. 5c, middle panel), we do not think that ZFP296 binding considerably affects the histone methylation of the upstream region of *Dppa3*. We agree that H3K9me2 and/or H3K9me3 ChIP-seq may provide important information, but considering the novel results based on ATAC-seq, we decided to reformulate our conclusions (see also Abstract). We removed the descriptions such as “ZFP296 plays essential roles in maintaining open chromatin structure in ESCs” from the whole manuscript.

Comment (2): ATAC-seq should be performed to support the authors statement that “ZFP296 is also considered to contribute to the maintenance of a generally open chromatin configuration ».

Response (2): According to the Reviewer’s suggestion, we performed ATAC-seq and combined analysis of the ATAC-seq, ChIP-seq, and RNA-seq gave us very exciting results. We are very grateful to the Reviewer. As described in the revised manuscript, chromatin accessibility around ZFP296 binding peaks was maintained at various levels in WT ESCs, but a considerable fraction showed dramatically increased accessibility in KO ESCs, which was often accompanied by elevated transcription of the closest genes. Thus, ZFP296, rather than keeping chromatin open, seems to attenuate chromatin accessibility of the ZFP296 binding regions. These results are shown in new Figs. 5 and 6 and described in the Results section. The Discussion section has been revised accordingly.

Comment (3): The assertion that ZFP296 could inhibit the G9a/GLP heterodimerization must be checked in vitro using recombinant proteins. In addition, authors should cross ZFP296 and G9A/GLP ChIPseq data and compare the ZFP296 and G9a/GLP KO induced phenotypes in ESCs.

Response (3): We are very interested in the relation between ZFP296 and G9a/GLP but would like to pursue it in our future research. We removed the last two sentences of the Results section describing possible inhibition of G9a/GLP heterodimerization by ZFP296.

Comment (4): Line 315: “The results showed a significant increase of H3K9me2 and me3, but not H3K9me1 or K27me3, in KO ESCs, suggesting that ZFP296 negatively regulates H3K9me2/3 levels”: at least discuss how these data fit with the lack of any difference in 5mc levels and an increase in 5hmc in KO cells (Supplemental Fig 5).

Response (4): It is known that the DNA and histone lysine methylation systems are highly interrelated. However, DNA methylation is regulated by various factors, such as DNA methyltransferases and Tet proteins. Expression levels of the genes coding these proteins also

seemed affected by *Zfp296* KO. Thus, it seems difficult to discuss on the relation between DNA and histone methylation.

Comment (5): To gain more insights into ZFP296 direct versus indirect gene targets cross the RNAseq and CHIPseq data.

Response (5): As described in our response to the comment (2) by the Reviewer 1, the systematical analyses of the results of RNA-seq, ChIP-seq, and ATAC-seq have been conducted by Drs. H. Yamano and D. Motooka who are specialized in this field and joined the present study as coauthors. The results are mainly shown in revised Figs. 2, 5, and 6. The Results and Discussion sections have been extensively revised to include these results.

SPECIFIC COMMENTS:

Comment (1): Fig 3b: “The percentage of beating EBs was considerably lower in Rescue ESCs, compared with the other cell lines during the observation period...”: this result seems surprising, where the WT and KO are comparable while the rescue behaves differently for WT and KO! The authors conclusion that “... suggesting that persistent expression of *Zfp296* suppresses mesodermal differentiation” is not satisfactory at all. Please comment.

Response (1): Once WT ESCs begin to differentiate, *Zfp296* expression declines rapidly, as shown in Supplementary Fig. 1a. WT and KO ESCs differentiated into beating EBs in a similar pattern. Considering that the expression of *Zfp296* severely affects the gene expression pattern (Fig. 2) and chromatin accessibility (Fig. 6), we think that persistent expression of *Zfp296* may possibly be harmful to mesodermal differentiation. We have changed the indicated sentence to “suggesting that persistent expression of *Zfp296* may exert inhibitory effects on mesodermal differentiation”.

Comment (2): Lines 106-108: “Interestingly, *Zfp296*-overexpressing ESCs formed homogeneous colonies with compacted morphology”: please cite a reference.

Response (2): Because *Zfp296* expression in KO ESCs gave the same result in regard to the colony morphology, we deleted this sentence and moved the description on the colony morphology (9 lines) under the next section “**Characterization of *Zfp296*-deficient ESCs**”.

Comment (3): Line 126: show the karyotype data.

Response (3): The karyotype data are as shown below.

KO#98 : 40, 40, 40, 40, 40, 40

KO#156 : 40, 39, 40, 40, 39, 40, 40, 40, 40, 39, 40, 40, 40, 40

We think that the ratio of normal karyotype would give sufficient information.

Comment (4): Lines 203-204 (and elsewhere): “The genes whose expression was the most severely affected are shown in Fig. 2a », please state the chosen thresholds and p-values instead of “most severely affected”.

Response (4): The conditions of filtering out genes whose expression was severely affected in RNA-seq are as shown below. Genes whose normalized read counts were less than 8 in all the ESCs and noncoding genes were omitted. Difference vs. KO: $P < 0.05$. These thresholds were described in the legend to Fig. 2.

Left panel shows the genes : WT > KO, more than 5.6-fold difference.

Right panel shows the genes: WT < KO, more than 3.2-fold difference.

Comment (5): Figure issues: Fig 1b and 1c, improve the Y-axis legend (proliferation and apoptosis (%) are not informative enough).

Response (5): We have changed the Y-axis legend in revised Fig. 1b and 1c.

Comment (6): Many figures do not contain any statistical analyses of the data unless I missed them (Figures 1d, 2 a-b-c, all Supplemental Fig 3). A “Statistical analyses” chapter must be included in the Methods section.

Response (6): P values have been added in Fig. 1d. Gene lists in Fig. 2a were obtained by statistical processing using EdgeR as described in Response (4). Since the experiment in Supplementary Fig 3a was conducted in technical triplicates, statistical processing results in significant differences in any combination of the cell lines for each gene. Because the purpose of this analysis is to detect any tendency of differentiation potential among WT, KO, and Rescue ESCs, we did not show the significance levels of differences in this figure. Statistical analysis in Supplementary Fig. 3b was performed by Tukey’s test between the three groups. A “**Statistics and reproducibility.**” subsection was added to the Methods section.

Comment (7): Fig 3a: IF quality does not meet publication standards. Show different enlargements, quantify IF signals and perform statistical analyses, comment the different DPP3 IF intensity signal within WT and rescue cells...

Response (7): We performed IF analysis of WT, KO, and Rescue ESCs with the DPPA3 antibody. As shown in new Fig. 3a, DPPA3 was clearly stained in WT and Rescue ESCs. Enlarged figures were also included. IF signal intensity did not seem considerably different between WT and Rescue ESCs. Since we are not familiar with the statistical analysis based on the fluorescence intensity of individual cells, western blotting was quantitatively performed as shown in Fig. 3b, comparing the amount of DPPA3 among the three groups (see below).

Comment (8): Fig 3b: western blot (WB) loading control, signal quantification and the number of replicates are missing. WB signals must be quantified in at least 3 independent experiments and statistically analyzed. The number of experiments must be stated in the figure legend.

Response (8): We performed the western blotting experiments using three biological replicates and the results are shown in revised Fig.3b. Lamin B1 served as the loading control. The relative intensity of the DPPA3 band to the Lamin B1 band of each lane was calculated using the Image Lab software. The bar graph shows relative values when WT intensity was set to 1.

Comment (9): Fig 5 a-b: confirm the RNAseq data by RT-qPCR. Improve the Y-axis scale form.

Response (9): Expression levels of the genes up- and down-regulated upon *Zfp296* KO including the ones shown in new Fig.5c-d were confirmed by RT-qPCR, and the results are shown in revised Supplementary Fig. 4. We have revised the Y-axis to clarify the scale.

Comment (10): Fig 5c and Fig 6a: Y-axis legend is missing.

Response (10): We have added Y-axis legend on Fig. 5c (revised Supplementary Fig. 5) and Fig. 6a (revised Fig. 7a).

Comment (11): Fig 6a: show WB image examples.

Response (11): We have added WB image examples on revised Fig. 7a, right panel.

Comment (12): General: it is not correct to state “genes down- or up-regulated by ZFP296” when mentioning genes down or up-regulated upon ZFP296 KO. Please modify this all over the MS.

Response (12): We agree with the Reviewer’s comment. We have changed the wording “genes down- or up-regulated by ZFP296” to “genes down- or up-regulated upon *Zfp296* KO” through the whole manuscript.

Comment (13): The ChIPseq raw data deposition is not stated in the MS.

- Page 29: “Data availability : The data supporting the findings of this study are available from the corresponding author upon reasonable request”: this is somehow surprising since the authors are supposed to make available all their raw data.

Response (13): The raw data of all the NGS analyses have been deposited in the NCBI GEO as described above. The deposition information is described in the last part of the Methods section (“Data availability”). We have changed the indicated sentence to “The data supporting the

findings of this study are available from the corresponding author upon request”.

Comment (14): The discussion section is way too speculative, especially when it is based on the RNAseq data (regarding *Otx2* and *Gbx2* gene expression for example) without any confirmation. It should be reduced.

Response (14): The Discussion section has been revised to reduce the description on *Otx2* and *Gbx2*.

MINOR COMMENTS:

Comment (1): Fig 4e legend title “ZFP296 binding to heterochromatin” is not correct. An IF does not show binding but localization. Please modify.

Response (1): Based on the results of ChIP-seq, one of the binding motifs of Zfp296 is major satellite repeat, but we agree that IF only shows ZFP296 localization to heterochromatin. We have changed the Fig. 4e legend title to “Localization of ZFP296 to heterochromatin”

Comment (2): Supplemental Table 1: add a legend.

Response (2): The data corresponding to Supplementary Table 1 have been deposited in the NCBI GEO.

Comment (3): Line 201: state which phenotypes.

Response (3): We have added description of “unique phenotypes” as follows; “The above studies demonstrated that *Zfp296*-KO ESCs grew as flat colonies and could not form teratomas but could contribute to germline-competent chimeric mice.”

Comment (4): Write Zfp296 instead of ZFP296 when mentioning the protein.

Response (4): According to the rules by the Mouse Genome Informatics (MGI) and International Protein Nomenclature Guidelines, the gene name should be *Zfp296*, and its product should be ZFP296. We have checked the whole manuscript to follow these rules.

Comment (5): General: please show the color code meaning in the heatmap figures (2a, Supplemental 5b).

Response (5): Fig. 2a has been revised, and color code meaning in the heatmap is described in the figure legend. Heatmap figure in Supplementary Fig. 5b was removed in the revised manuscript to avoid too speculative discussion.

Reviewer #3:

Comment (1): The authors state in the title that “Zfp296-KO ESCs retain unique ground-state”, my impression is that absence of Zfp296 impairs differentiation rather than contributing to pluripotency. This would for example explain why the KO cells are unable to undergo teratoma formation. I would be therefore careful with the title.

Response (1): There may be opinions different from ours that *Zfp296*-KO ESCs retain ground state of pluripotency, because they cannot form teratomas. Considering the Reviewer’s another comment (8), we have changed the title to “**Zfp296 knockout enhances chromatin accessibility and induces a unique state of pluripotency in embryonic stem cells.**”

Comment (2): The changes in morphology are interesting and indeed reflect the differentiation status of ESC populations, but are not conclusive. *Zfp296*-overexpressing ESCs as well as rescued *Zfp296*^{-/-} cells form homogeneous colonies similar to the WT ESCs (Fig.1). Both WT and knockout ESCs are germline competent independent of their morphology, therefore I doubt that *Zfp296* has a role in maintaining pluripotency even if the morphology of the colonies is changing. A staining of the cells with markers like Oct-4, SSEA1, Rex1 and Nanog of the colonies could give some important additional information.

Response (2): We think that ZFP296 has a role in exit of ESCs from naïve pluripotent state. Among the indicated 4 markers, immunostaining of *Nanog* was performed, and we found no considerable difference in staining pattern among WT, KO, and Rescue cells (see below). Real-time PCR analysis of *Oct-4*, *Rex1* (*Zfp42*), and *Nanog* is shown in Supplementary Fig. 3. These pluripotency-related genes were expressed at high levels in WT, KO, and Rescue ESCs.

Comment (3): The fact that *Zfp296* KO ESCs are not able to form teratoma is interesting, but at least for mouse ESCs the proof of pluripotency is the capacity to form germline competent chimera, and the authors show that the absence of *Zfp296* does not impair this capacity. Also here, the interpretation could be that *Zfp296* is not necessary for keeping a pluripotent state. In this context, how does the stable expression of *Zfp296* (expression that cannot be downregulated upon injection in the blastocyst, e.g. rescue lines) influence pre-/post-

implantation embryonic development? Or, at least are Zfp296 KO cells able to differentiate in vitro?

Response (3): We have not examined whether Rescue ESCs can form chimeric mice.

Supplementary Fig. 3 shows changes in gene expression and the number of beating EBs when cultured without LIF. Differentiation into beating EBs was considerably inhibited in Rescue cells. Therefore, we think that persistent expression of *Zfp296* is probably harmful to the embryonic development.

Comment (4): Is the expression of Zfp296 in LIF/serum similar to 2i/LIF (ground state)?

Response (4): *Zfp296* expression under the 2i/LIF condition is interesting to us, but because all the experiments of this study were conducted under the LIF/serum condition, we do not know the expression level of *Zfp296* under the 2i/LIF condition.

Comment (5): Open chromatin is a characteristic of pluripotent cells but it is not a determinant of pluripotency. The data presented highlighting the role of Zfp296 in maintaining an open chromatin structure are well described. But if this is the reason why Zfp296 KO ESCs retain unique ground-state pluripotency cannot be determined by the experiments presented in the manuscript.

How is the methylation changing upon differentiation of Zfp296^{-/-} cells in vitro? Are the cells able to terminally differentiate? Can they be upon differentiation brought back to a pluripotent state upon addition of serum/LIF?

Response (5): According to the Reviewer 2's comment, we performed ATAC-seq. Combined analysis of ATAC-seq, ChIP-seq, and RNA-seq gave us unexpected results. As described in the revised manuscript, chromatin accessibility around ZFP296 binding peaks was maintained at various levels in WT ESCs, but a considerable fraction showed dramatically increased accessibility in KO ESCs, which was often accompanied by elevated transcription of the closest genes (see new Figs. 5 and 6). Thus, ZFP296, rather than keeping chromatin open, seems to restrict chromatin accessibility of its binding regions. The Results and Discussion sections have been extensively revised accordingly. We did not pursue methylation changes in KO ESCs during differentiation. In vitro differentiation seemed to proceed normally in the absence of ZFP296 (Supplementary Fig. 3). We have not examined if differentiated KO ESCs can be brought back to a pluripotent state upon addition of serum/LIF.

Comment (6): A more general comment that should be addressed in the discussion: It is not clear to me how is it possible that Zfp296 knockout ESCs are able to generate germline competent chimeric animals when Zfp296 was previously shown to be responsible for

acquisition of PGC fate and its abrogation leading to failure in activating germline genes and inducing loss of germ cell identity. Are here compensatory effects playing a role?

Response (6): In our previous paper on *Zfp296*-KO mice (Matsuura et al. 2017), we showed that PGCs are formed in homozygous KO mice but much less than in WT mice and a small percent of KO male mice were fertile. In chimeric mice, presence of WT cells might have caused some compensatory effects on the PGC development from KO cells, but we have not examined this possibility.

Comment (7): In conclusion, the data presented is very interesting but not conclusive. Despite the changes in morphology, the inability to form teratoma and the increase in histone H3K9 di- and trimethylation *Zfp296* KO cells retain a pluripotent state and generate germline competent chimera. The data can be interpreted also in the direction that *Zfp296* is expressed in ESCs and its depletion does not impair their pluripotent state, means *Zfp296* is not important for self-renewal and pluripotency. I therefore would strongly recommend the authors not to use the concept of retaining “unique ground-state” pluripotency in the context of *Zfp296* KO cells but rather state that *Zfp296* cells are pluripotent.

Response (7): Interpretation of the pluripotency state of *Zfp296*-KO ESCs seems complicated. According to the Reviewer’s comment, we have changed the title as described in our Response (1). In addition, we removed descriptions such as ground-state pluripotency in the context of *Zfp296*-KO ESCs.

REVIEWERS' COMMENTS:

Reviewer #1 (Remarks to the Author):

The quality of revised manuscript has been significantly promoted and the authors have addressed most of my concerns. However, some of the statements in this manuscript were overstated. For example, line362-363 "clearly demonstrated". The authors should be careful to use "clearly" in the whole manuscript.

Reviewer #2 (Remarks to the Author):

The authors have largely addressed my main concerns.

Reviewer #3 (Remarks to the Author):

I have no additional comment to the authors.

The only comment we should respond was the following one.

Comment by Reviewer #1: Some of the statements in this manuscript were overstated. For example, line362-363 "clearly demonstrated". The authors should be careful to use "clearly" in the whole manuscript.

We agree with this comment and removed the word “clearly” in lines 234, 362, 381, 465, and 525 in the manuscript.